# Recycled melanoma-secreted melanosomes regulate tumor-associated macrophage diversification

Roma Parikh [1], Shivang Parikh [1,16], Daniella Berzin[2], Hananya Vaknine[3], Shai Ovadia [4], Daniela Likonen[2], Shoshana Greenberger [2], Alon Scope [5], Sharona Elgavish[6], Yuval Nevo[6], Inbar Plaschkes[6], Eran Nizri[7,8], Oren Kobiler [9], Avishai Maliah[1], Laureen Zaremba[10], Vishnu Mohan [11], Irit Sagi[11], Ruth Ashery-Padan[4], Yaron Carmi[12], Chen Luxenburg [13], Jörg D Hoheisel [10], Mehdi Khaled[14], Mitchell P Levesque [15] & Carmit Levy [1✉]

## Abstract

Extracellular vesicles (EVs) are important mediators of communication between cells. Here, we reveal a new mode of intercellular communication by melanosomes, large EVs secreted by melanocytes for melanin transport. Unlike small EVs, which are disintegrated within the receiver cell, melanosomes stay intact within them, gain a unique protein signature, and can then be further transferred to another cell as "second-hand" EVs. We show that melanoma-secreted melanosomes passaged through epidermal keratinocytes or dermal fibroblasts can be further engulfed by resident macrophages. This process leads to macrophage polarization into pro-tumor or pro-immune cell infiltration phenotypes. Melanosomes that are transferred through fibroblasts can carry AKT1, which induces VEGF secretion from macrophages in an mTOR-dependent manner, promoting angiogenesis and metastasis in vivo. In melanoma patients, macrophages that are co-localized with AKT1 are correlated with disease aggressiveness, and immunotherapy non-responders are enriched in macrophages containing melanosome markers. Our findings suggest that interactions mediated by second-hand extracellular vesicles contribute to the formation of the metastatic niche, and that blocking the melanosome cues of macrophage diversification could be helpful in halting melanoma progression.

**Keywords** Angiogenesis; Cell-to-Cell-Transfer; Heterogeneity; Melanosomes; Tumor Associated Macrophages
**Subject Categories** Cancer; Immunology; Membranes & Trafficking

## Introduction

Melanoma, a melanocyte-origin neoplasm, is the most lethal of skin cancers, with an incidence of about 100,000 new cases annually. Despite significant strides made in melanoma immunotherapy, a minority of patients respond to treatment, and disease recurrence often ensues, posing a significant clinical challenge (Pittet et al, 2022). Investigation of the melanoma microenvironment and study of how cancerous cells communicate with neighboring cells is pivotal to the identification of new therapeutic targets (Falcone et al, 2020; Wang et al, 2022).

Extracellular vesicles (EVs) refer to a group of membrane-bound vesicles generated by cells through diverse mechanisms including outward budding from the plasma membrane or inward budding of the endosomal membrane (van Niel et al, 2022). EVs are key players in cellular communication networks, including the communication between melanoma cells and their microenvironment (Dror et al, 2016). Melanosomes, which are synthesized specifically in melanocytes, originate from the early endosomal system (Raposo and Marks, 2007) and are tissue-specific, membrane-bound organelles of 200–500 nm in size (Hearing, 2005; Raposo and Marks, 2002) that transport melanin pigment to neighboring epidermal keratinocytes as protection against UV-induced DNA damage (Raposo and Marks, 2007; Yamaguchi and Hearing, 2009; Joshi et al, 2007). Although melanosomes are secreted out of the cell of origin and meet other criteria that could allow them to be classified as EVs (according to the Minimal Information for Studies of Extracellular Vesicles (Théry et al, 2018), including their size and inclusion of a cargo of RNA and protein content, they are not classical EVs at this stage as they are secreted without a membrane and thus have been termed melanocores (Théry et al, 2018). However, melanocores gain a

[1]Department of Human Genetics and Biochemistry, Sackler Faculty of Medicine, Tel Aviv University, Tel Aviv 69978, Israel. [2]Institute of Pathology, Sheba Medical Center, Tel Hashomer 52621, Israel. [3]Institute of Pathology, E. Wolfson Medical Center, Holon 58100, Israel. [4]Department of Human Molecular Genetics and Biochemistry, Sackler Faculty of Medicine and Sagol School of Neurosciences, Tel Aviv University, Tel Aviv 69978, Israel. [5]The Kittner Skin Cancer Screening and Research Institute, Sheba Medical Center and Sackler Faculty of Medicine, Tel Aviv University, Tel Aviv, Israel. [6]Info-CORE, Bioinformatics Unit of the I-CORE at the Hebrew University of Jerusalem and Hadassah Medical Center, Jerusalem 91120, Israel. [7]Department of Dermatology, Tel Aviv Sourasky (Ichilov) Medical Center, Tel Aviv 6423906, Israel. [8]Sackler School of Medicine, Tel Aviv University, Tel Aviv 69978, Israel. [9]Department of Clinical Microbiology and Immunology, Sackler School of Medicine, Tel Aviv Universitygrid.12136.37, Tel Aviv, Israel. [10]Division of Functional Genome Analysis, German Cancer Research Center (DKFZ), Heidelberg, Germany. [11]Department of Biological Regulation, Weizmann Institute of Science, Rehovot 7610001, Israel. [12]Department of Pathology, Sackler School of Medicine, Tel Aviv University, Tel Aviv 69978, Israel. [13]Cell and Developmental Biology, Sackler School of Medicine, Tel Aviv University, Tel Aviv 69978, Israel. [14]INSERM 1279, Gustave Roussy, Université Paris-Saclay, Villejuif, France. [15]Department of Dermatology, University of Zurich, University Hospital Zurich, Wagistrasse 18, CH-8952 Schlieren, Switzerland. [16]Present address: The Ragon Institute of Mass General, Massachusetts Institute of Technology (MIT), and Harvard, MA 02139 Cambridge, USA. ✉E-mail: carmitlevy@post.tau.ac.il

membrane once inside keratinocytes, resulting in novel membrane-bound organelles (Tarafder et al, 2014; Hurbain et al, 2018; Benito-Martínez et al, 2021). Further, there are additional ways of melanosome secretion, including the shedding of large vesicles that contain melanosomes with their membrane (Théry et al, 2018; Benito-Martínez et al, 2021). Melanoma cells retain the capacity to produce melanosomes (Basrur et al, 2003; Lazova and Pawelek, 2009). Although the reasons for this are largely unknown, evidence suggests that melanosomes play more complex roles in the progression of melanoma and in response to treatment than previously thought. First, melanosome production demonstrates a plasticity phenotype that appears to correlate with the disease stage (Pinner et al, 2009). Second, these EVs are produced and secreted at metastatic locations (Netanely et al, 2021). Third, the uptake of melanosomes into dermal fibroblasts transforms these cells into cancer-associated fibroblasts (CAFs) (Dror et al, 2016), and melanosomes induce pro-lymphangiogenic changes in dermal lymphatic cells to promote metastatic niche generation (Leichner et al, 2023). Fourth, patients with pigmented melanoma have poorer survival than those with amelanotic melanoma (Brożyna et al, 2013). Finally, melanogenesis impairs the sensitivity of melanoma cells to radio-(Brozyna et al, 2008), chemo-, and immunotherapy (Slominski et al, 2009), and blocking of melanogenesis enhances the response to these therapies (Brozyna et al, 2008; Slominski et al, 2009), indicating that melanosomes influence the immune response to melanoma.

Macrophages are the most abundant immune cells within tumors. Tumor-associated macrophages (TAMs) influence tumor progression and interfere with cancer therapy (DeNardo and Ruffell, 2019; Ma et al, 2022). TAM heterogeneity is a major obstacle to the study of their functions (Ma et al, 2022; Pittet et al, 2022). This heterogeneity is reflected in the contradictory roles attributed to TAMs during melanoma progression (Ma et al, 2022; Pittet et al, 2022). TAMs formed from circulating monocytes promote cancer progression by enhancing tumor growth, invasion, metastasis, and the release of the growth factor VEGF (Peranzoni et al, 2018), which is necessary for the vascularization and evasion of immune cells (Pieniazek et al, 2018; Pittet et al, 2022). On the other hand, TAMs can kill and phagocytose cancer cells (Evans and Alexander, 1970) and are known to promote immune activity in the tumor microenvironment (Oshi et al, 2020), which are anti-tumor functions.

TAM heterogeneity has made generation of an accurate definition of the TAM population difficult. Over a decade ago, it was proposed that, based on environmental cues, macrophages are polarized into the simple dichotomy of the M1 and M2 phenotypes (Pittet et al, 2022; Ma et al, 2022). The M1 population, induced by toll-like receptors and type 1 cytokines such as IFNγ and TNFα, are pro-inflammatory cleaners (Ma et al, 2022; Lu et al, 2018). The M2 population, which have anti-inflammatory or remodeler roles, are induced by IL-4, IL13, or TGFβ and other profibrotic factors such as PDGF (Ma et al, 2022; Lu et al, 2018). However, recent work has shown that TAMs exhibit a range of activation states that do not fit neatly into M1/M2 model (Pittet et al, 2022; Martinez and Gordon, 2014; Nahrendorf and Swirski, 2016). Characterization by cellular indexing of transcriptomes and epitopes by sequencing that combines single-cell RNA sequencing and proteomics suggests that there are multiple TAM subtypes (Ma et al, 2022). TAM classification is an active area of research, and the cues that induce the formation of various subtypes are largely known.

Macrophage that take up melanosomes are known as melano-phages (Pawelek, 2007; Handerson et al, 2007). In electron microscopy images of melanoma specimens, we observed that melanosomes taken up by epidermal keratinocytes and epidermal fibroblasts that neighbor melanoma cells remain intact. We hypothesized that keratinocytes and fibroblast might also secrete melanosomes and that macrophages, the "cleaner" cells of the skin, may take up melanosomes from these cells as well as melanosomes secreted by melanoma cells. Here, we report an examination of this hypothesis. Using a melanoma cell line that stably expresses labeled melanosomes, we followed melanosomes secreted by melanoma cells into keratinocytes and fibroblasts and showed that these EVs can be further transferred into macrophages. Surprisingly, macrophages that had been cultured with melanosomes from different cell sources had different effects on tumor progression in mice. Macrophages that had taken up fibroblast melanosomes enhanced tumor growth, vascularization, and metastasis, whereas macrophages that had been cultured with keratinocyte-derived melanosomes enhanced immune infiltration. These phenotypic differences were also detected in transcriptomic analyses. Proteomic analyses of the melanosomes revealed a protein signature that was unique to each host cell. Fibroblast-derived melanosomes contained AKT1, not present in melanosomes that originated from melanoma cells. The presence of AKT1 polarized macrophages toward a pro-tumor phenotype. Blocking the activity of a downstream target of AKT1, mTOR, using rapamycin, inhibited the pro-tumor effects of the fibroblast-derived melanosomes on macrophages. Supporting the clinical relevance of fibroblast-derived macrophages, enrichment of downstream targets of AKT correlated with clinical melanoma progression.

# Results

## Melanosomes are detected in non-cancerous cells in the tumor microenvironment

To investigate the presence of macrophages in the microenvironment during melanoma development, we analyzed pathological specimens of compound nevi, melanoma in situ, vertical melanoma, and lymph node metastases. We observed large cells with ill-defined cytoplasmic borders filled with melanin pigment in all specimens (Figs. 1A and EV1A). Cells of this description were previously termed melanophages (Busam et al, 2001). To validate that these cells were indeed melanophages, we immunostained three independent patient specimens with a melanoma marker HMB45 (Golan et al, 2019), the mature melanosome marker GPNMB (Dror et al, 2016), and a pan-macrophage marker CD68 (Pittet et al, 2022). Cells that co-stained with CD68 and GPNMB were scattered in the compound nevi, whereas in the in situ and vertical melanomas and lymph-metastasized melanoma, these co-stained cells were located internal to the melanoma cells or within the tumor microenvironment (Figs. 1A,B and EV1A). Taken together, our results show that macrophages take up melanosomes in all stages of melanoma development. As expected based on previous studies (Yamaguchi and Hearing, 2009; Joshi et al, 2007; Dror et al, 2016), the epidermis and dermis of the skin of patient specimens contained keratinocytes and fibroblasts, respectively, and these cells also contained melanosomes (Fig. EV1B).

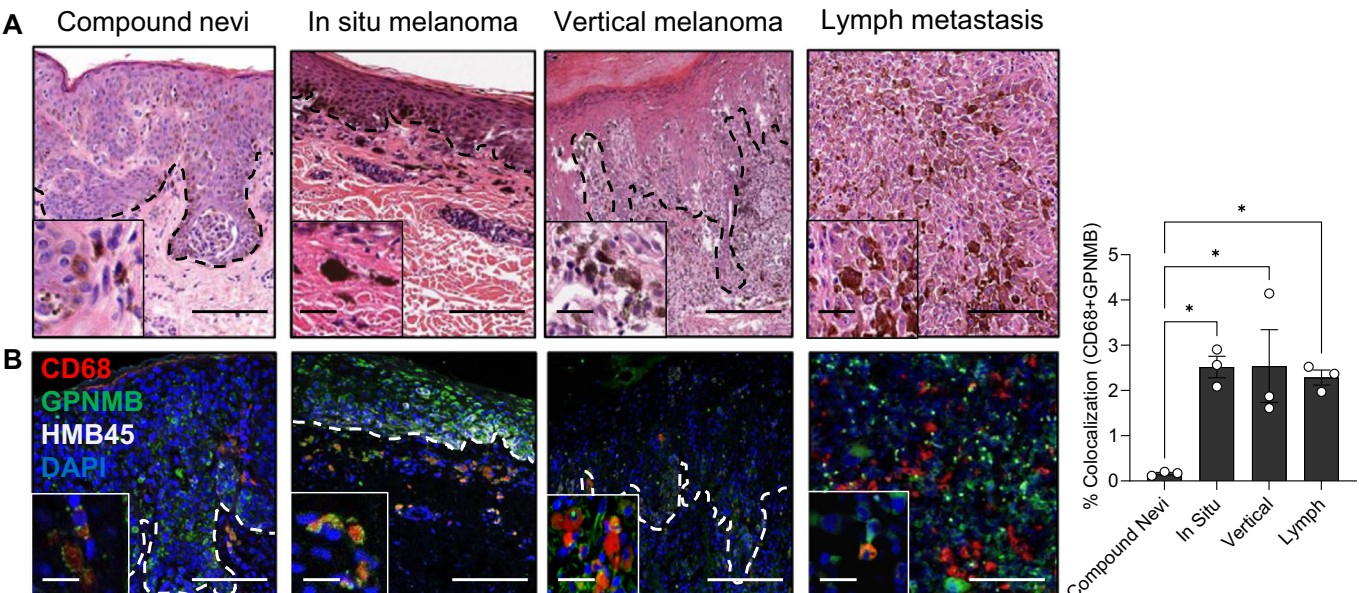

**Figure 1. Melanosomes are detected in non-cancerous cells in the tumor microenvironment.**

(A) Representative images of cross-sections of compound nevi, melanoma in situ, vertical melanoma, and lymph metastasis stained with hematoxylin and eosin (H&E). Black dashed lines demarcate the epidermal from the dermal regions. Scale bars, 200 μm. Inset images show macrophages containing pigmented vesicles. Scale bars, 50 μm. (B) Consecutive slices of specimens shown in panel A immunostained for HMB45 (white), melanosomal marker GPNMB (green), and macrophage marker CD68 (red). Nuclei are stained with DAPI (blue). White dashed lines demarcate the epidermal from the dermal regions. Scale bars, 200 μm. Inset images show the co-localization of macrophages with melanosomes. Scale bars, 50 μm. The graph depicts the percentage co-localization of macrophages (CD68) with melanosomes (GPNMB) in melanoma cross-sections as analyzed using ImageJ. $n = 3$ patients from each stage of melanoma cross-sections. Data information: In (B), error bars represent ±SEM. One-way ANOVA was performed for statistical analysis; *$P \le 0.05$ was considered significant. Source data are available online for this figure.

## Cell-to-cell transfer of melanoma cell-derived melanosomes occurs in the tumor microenvironment

To study the uptake of melanosomes by keratinocytes, fibroblasts, and macrophages, we isolated melanosomes secreted by melanoma cells. The melanosome preparation was shown to be free of small vesicles by NanoSight and transmission electron microscopy (TEM) analyses, which revealed the characteristic size range (200–500 nm) and dark elliptical morphology of the secreted melanosomes (Dror et al, 2016) (Fig. EV2A). These melanosomes were cultured with HaCaT cells, which are immortalized keratinocytes, human primary fibroblasts, or human primary monocyte-derived macrophages. After 24 h of culture with melanosomes, we observed relatively spherical organelles (200–500 nm) containing dark vesicles (Correia et al, 2018; Hurbain et al, 2018) within the cytoplasm's of keratinocytes and fibroblasts by TEM and immuno-fluorescence analyses (Figs. 2A,B and EV2B). MNT1 melanoma cells were used as a positive control. Moreover, we demonstrate that there is a clear separation between the melanosome organelle and the lysosomes surrounded by the multi-membrane sheets within the keratinocytes (Fig. 2B). Interestingly, although melanosomes were intact in keratinocytes and fibroblasts, within the macrophages we detected broken melanosomes in a multi-membrane lysosome-like compartment (Fig. 2B).

To investigate whether melanophages take up melanosomes secreted by melanoma cells or whether the melanosomes observed in melanophages are derived from keratinocytes or fibroblasts, we first validated that melanosome stays intact in keratinocytes and

fibroblasts. From MNT1 melanoma cells, we isolated melanosomes and also small EVs (50–200 nm), as validated using the TEM and NanoSight analysis (Fig. EV2A,C). MISEV guidelines suggest that a preliminary step in functional studies should involve comparing the activity of small EVs with that of large EVs (Théry et al, 2018) and given the size of melanosomes (resemble of large EVs), we used small EVs as a comparison. Moreover, the small EVs were used as a control for a vesicle known to be dissolved in the recipient cell (Gurung et al, 2021). Keratinocytes, fibroblasts, and macrophages were treated with membrane-labeled melanosomes and small EVs followed by fluorescence analysis for the following 3 days (Fig. 2C). The fluorescence signal due to melanosomes remained high in keratinocytes and fibroblasts for 2- and 3-days post-treatment, respectively; in contrast, little signal due to small EVs was observed on the second day (Fig. 2D, left and middle panel). In macrophages, fluorescence signals due to both melanosomes and small EVs were decreased significantly on day 2 (Fig. 2D, right panel), confirming our TEM observation that potential degradation of melanosomes begins at 24 h post uptake.

We observed pigmented pellets in the conditioned media of the keratinocytes and fibroblasts (Fig. 2E), suggesting that these cells secrete melanosomes, providing a reservoir of melanosomes that could be taken up by macrophages. Conditioned media from MNT1 and HeLa cells were used as positive and negative controls for secreted melanosomes, respectively (Fig. 2E). No melanosome pellet was observed in conditioned media from the macrophages (Fig. 2E), supporting our hypothesis that melanosomes are degraded in macrophages. TEM analysis of melanosomes isolated

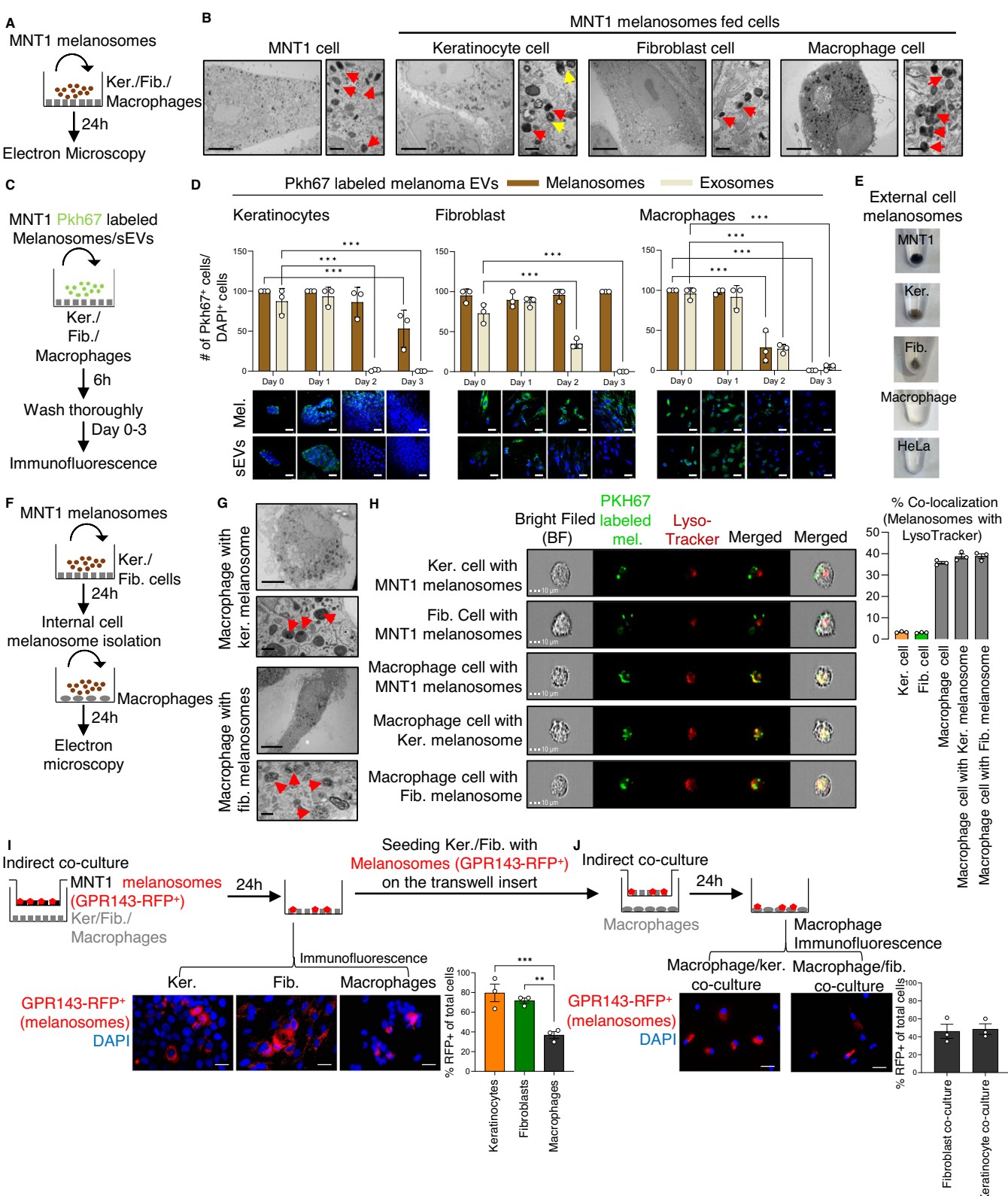

**Figure 2. Cell-to-cell transfer of melanoma cell-derived melanosomes occurs in the tumor microenvironment.**

(A) Workflow for the culture of melanosomes secreted by MNT1 cells with keratinocytes (Ker.), fibroblasts (Fib.), or naïve macrophages. (B) TEM images of keratinocytes, fibroblasts, and macrophages cultured with melanosomes secreted by MNT1 cells for 24 h. Scale bars, 10 μm. Enlarged images show melanosomes (indicated by red arrowheads) within the cytoplasm of the cells. Yellow arrowheads indicate lysosomes. Scale bars, 0.5 μm. (C) Workflow for labeling of melanosomes (Mel.) and small EVs (sEVs) from MNT1 melanoma cells with Pkh67 dye and subsequent culture of keratinocytes, fibroblasts, or naïve macrophages with EVs followed by a thorough washing of the cells. (D) Upper: Percentage of PKH67-labeled cells of all DAPI-labeled cells in cultures of keratinocytes (left panel), fibroblasts (middle panel), and macrophages (right panel) cultured with Pkh67-labeled melanosomes or small EVs. $n = 4$ independent experiments. Lower: Representative immunofluorescence images of indicated cells cultured with Pkh67-labeled melanoma EVs (green). DAPI-stained nuclei appear blue. Scale bars, 100 μm. (E) Representative photographs of melanosomal pellets from conditioned media of cells cultured for 3 days with Pkh67-labeled melanoma EVs. MNT1 and HeLa cells were used as positive and negative controls, respectively. (F) Workflow for isolation of melanosomes internalized by keratinocytes or fibroblasts and subsequent culture with macrophages. Cells were cultured with MNT1 melanosomes for 24 h and homogenized. Extracts were separated by sucrose gradient centrifugation, and the melanosome fraction was collected. Naïve macrophages were cultured with the isolated melanosomes for 24 h and then analyzed using TEM. (G) TEM images of macrophages cultured with melanosomes isolated from keratinocytes (upper panel) and from fibroblasts (lower panel). Scale bars, 10 μm. Enlarged images show broken melanosomes (indicated by red arrowheads) within the cytoplasm of the cells. Scale bars, 0.5 μm. (H) Representative ImageStream cytometry showing images of single keratinocytes, fibroblasts, and macrophages that have taken up PKH67-labeled melanosomes. Cells were stained with the LysoTraker Deep red dye. Scale bar = 10 μm. The graph depicts the percentage co-localization of the melanosomes with the LysoTracker dye in each condition. $n = 3$ independent experiments. (I) Upper: Workflow of co-culture assay in which MNT1 cells stably expressing GPR143-RFP were co-cultured in a transwell system with keratinocytes, fibroblasts, or naïve macrophages for 24 h. Lower left: Representative images of indicated cells cultured with MNT1 cells stably expressing GPR143-RFP (red). DAPI-stained nuclei appear blue. Scale bars, 20 μm. Lower right: Percentage of cells positive for the RFP normalized to the number of DAPI-positive cells. $n = 3$. (J) Upper: Workflow for co-culture of keratinocytes or fibroblasts, which had been previously co-cultured with melanoma cells with RFP-labeled melanosomes, with naïve macrophages. Lower left: Representative images of macrophages cultured with keratinocytes or fibroblasts previously co-cultured with melanoma cells expressing GPR143-RFP (red). DAPI-stained nuclei appear blue. Scale bar, 20 μm. Lower right: Percentage of cells positive for the RFP normalized to the number of DAPI-positive cells. $n = 3$. Data information: In panel (D, H–J), error bars represent ±SEM. One-way ANOVA was used for statistical analysis; *$P \leq 0.05$, **$P \leq 0.01$, ***$P \leq 0.001$ was considered significant. Source data are available online for this figure.

from keratinocytes and fibroblasts at 72 h post melanosome uptake and of melanosomes secreted by these cells revealed the classic morphology of melanosomes (Fig. EV2E). This indicates that melanosomes taken up by keratinocytes and fibroblasts in the melanoma microenvironment stay intact within the cells. These melanosomes could be secreted and taken up by macrophages, in a cell-to-cell transfer of EVs mechanism.

To investigate the hypothesis that macrophages can take up melanosomes that originate from keratinocytes and fibroblasts, we analyzed the EVs that had been internalized in these cells (Fig. 2F,G). TEM and NanoSight analyses of the isolated melanosomes revealed that these EVs had the classic morphology and size of the melanosomes (Fig. EV2F,G). TEM analysis revealed that, like melanosomes from melanoma cells, melanosomes from keratinocytes and fibroblasts were degraded after uptake into macrophages (Fig. 2G). As melanosomes function in melanin pigment synthesis and storage (Raposo and Marks, 2007), we analyzed their melanin content. High melanin concentrations were found in the EV fractions of both internalized and secreted melanosomes from the different source cells, whereas melanin was not present in the fractions containing small EVs (Fig. EV2H). Further, we detected signature melanosomal markers TYR, DCT, GPR143, and GPNMB (Dror et al, 2016; Raposo and Marks, 2007; Giordano et al, 2009; Hoashi et al, 2010) in the melanosomes derived from melanoma cells, fibroblasts and keratinocytes (Fig. EV2I). Moreover, we also detected the signature small EV proteins, including CD9, CD63, and Syntenin1 (Théry et al, 2018; Verweij et al, 2018) in the small EV fraction from melanoma cells, fibroblasts, and keratinocytes (Fig. EV2J). To validate the degradation of the melanosomes within the macrophages, we stained keratinocytes, fibroblasts and macrophages that were cultured with Pkh67-labeled melanosomes for 24 h with the LysoTracker dye [for acidic organelles such as lysosomes or autolysosomes (Raben et al, 2009; Correia et al, 2018)]. Analyses at a single-cell resolution showed that nearly 70% of keratinocytes, fibroblasts, and macrophages were positive for both the LysoTracker and Pkh67-labeled

melanosomes (Appendix Fig. S1A–F). In about 40% of macrophages, there was co-localization between melanosomes and the LysoTracker dye, whereas co-localization was observed in only 2–3% of fibroblasts and keratinocytes (Fig. 2H). These observations indicate that melanoma-secreted melanosomes stay intact or are degraded in a cell-dependent manner.

Next, we used MNT1 melanoma cells that stably express melanosomal protein GPR143 fused to the red fluorescent protein RFP (Adelmann et al, 2020) to track melanosome movement. When keratinocytes, fibroblasts, or macrophages were co-cultured with MNT1-GPR143 cells for 24 h, an RFP signal was detected in all three cell types (Fig. 2I). This indicates that melanosomes secreted by melanoma cells are transported into various cells in the tumor microenvironment including keratinocytes, fibroblasts, and macrophages, supporting previous findings (Yamaguchi and Hearing, 2009; Joshi et al, 2007; Dror et al, 2016; Pawelek, 2007). Keratinocytes and fibroblasts containing the MNT1 melanosomes were further co-cultured with naive macrophages. After 24 h of co-culture, RFP signal was detected in the macrophages (Fig. 2J), indicating that both keratinocytes and fibroblasts can transfer melanosomes to macrophages. Taken together, our findings suggest that melanosomes that originate in melanoma cells can be taken up by neighboring keratinocytes and fibroblasts and that melanosomes that have been resident in these cells can be transferred to macrophages.

## Macrophages cultured with melanosomes have heterogenous effects on cancer hallmarks that depend on the melanosome source

Macrophages in the tumor microenvironment exhibit significant heterogeneity, which likely accounts for their contradictory roles in promoting or suppressing tumors (DeNardo and Ruffell, 2019; Ma et al, 2022). The environmental cues that regulate diverse macrophage phenotypes are poorly understood. Evidence indicates that some melanophages support and some suppress tumor growth,

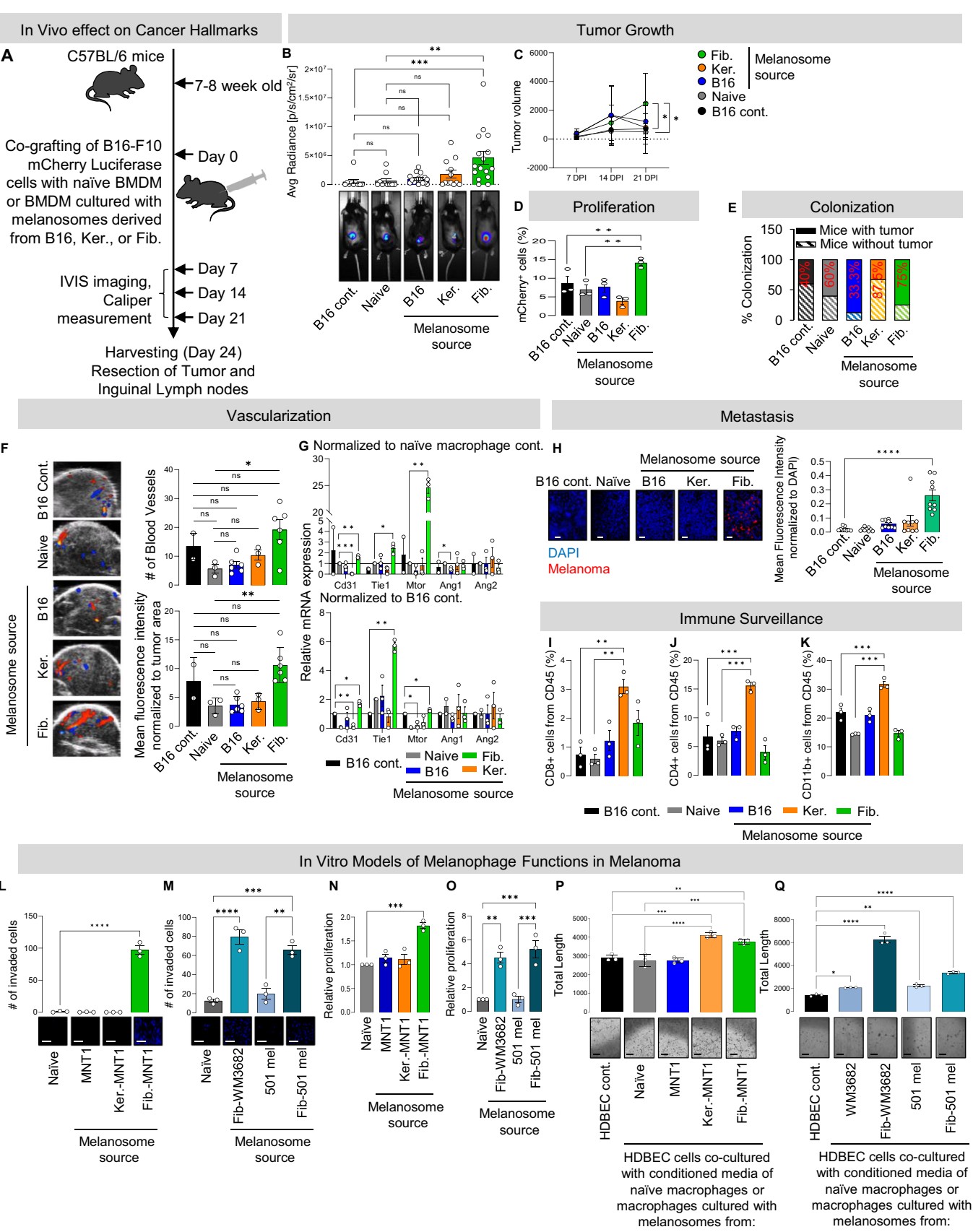

**In Vivo effect on Cancer Hallmarks**

**Tumor Growth**

**Proliferation**

**Colonization**

**Vascularization**

**Metastasis**

**Immune Surveillance**

**In Vitro Models of Melanophage Functions in Melanoma**

**Figure 3.   The effect of macrophages cultured with melanosomes on cancer hallmarks depends on the melanosome source.**

(A) Schematic workflow of the in vivo experiment. BMDMs were cultured with melanosomes from different sources. B16-F10 mouse melanoma cells that stably express mCherry and luciferase were grafted alone or co-grafted with naïve BMDMs or BMDMs cultured with melanosomes into C57BL/6 mice at a ratio of 1:5 (B) Upper: Average bioluminescence of each mouse on 7, 14, and 21 days post injection (DPI). Bars plot means of each group. $n \geq 5$ mice per group. Lower: Representative IVIS images of mice at 21 DPI. Note: The representative images shown here are re-used in Fig. EV3A as part of all the raw images. (C) Tumor volumes on 7, 14, and 21 DPI for mice from indicated groups. $n \geq 5$ mice per group. (D) Percent expression of mCherry-positive cells of total DAPI-labeled cells as determined by flow cytometry. $n = 3$ mice per group. (E) Percent of mice with tumors at 21 DPI. (F) Left: Representative color Doppler images highlighting vasculature in red and blue at 21 DPI. Upper right: Number of blood vessels per tumor. Lower right: Mean fluorescence intensity normalized to the area of the tumor. $n \geq 2$. (G) *Cd31, Tie1, Mtor, Ang1,* and *Ang2* mRNA levels in tumors from indicated groups relative to naïve BMDM control (upper panel) and relative to B16-F10 control (lower panel) at 24 DPI. $n = 3$ mice per group. (H) Left: Images of inguinal lymph nodes stained for melanoma cells (red) and nuclei (blue). Right: Mean fluorescence intensity of staining for melanoma marker normalized to DAPI. $n = 3$ mice per group and 3 images per mouse. (I) Percent CD8$^+$ T cells of total CD45$^+$ cells as analyzed using flow cytometry. $n = 3$ mice per group. (J) Percent CD4$^+$ T cells of total CD45$^+$ cells. $n = 3$ mice per group. (K) Percent CD11b$^+$ cells of total CD45$^+$ cells. $n = 3$ mice per group. (L) Upper: Numbers of invaded WM3682 melanoma cells. Melanoma cells were treated with conditioned media from naïve macrophages or macrophages cultured with melanosomes from indicated sources for 24 h. $n = 3$ independent experiments. Lower: Representative images of invasion assay. Scale bar, 50 μm. (M) Same as L but conditioned media from macrophages cultured with melanosomes from indicated sources. (N) Proliferation of WM3682 human melanoma cells treated with the conditioned media from naïve macrophages or macrophages cultured with melanosomes from indicated sources for 24 h. $n = 3$ independent experiments. (O) Same as N but conditioned media from macrophages cultured with melanosomes from indicated sources. (P) Total tube lengths in cultures of HDBECs incubated with conditioned media from naïve macrophages or macrophages cultured with melanosomes from indicated sources for 24 h. $n = 3$ independent experiments with cells from independent donors. (Q) Same as P but conditioned media from macrophages cultured with melanosomes from indicated sources. Note: The representative image of the HDBEC control (cont.) has been re-used in Fig. 5G as part of the same experiment. Data information: In panel (B–D, F–Q), error bars represent ±SEM. One-way ANOVA used for statistical analysis; *$P \leq 0.05$, **$P \leq 0.01$, ***$P \leq 0.001$, ****$P \leq 0.0001$ was considered significant. Source data are available online for this figure.

but the mechanistic explanation for this remains unknown (Rachkovsky et al, 1998; Handerson et al, 2007; Moro et al, 2021). As our data indicated that macrophages can take up melanosomes from different cell types in the tumor microenvironment, we sought to examine the effects of melanosome cell sources on melanoma cancerous hallmarks (Hanahan, 2022; Pittet et al, 2022). To this end, melanosomes secreted by B16-F10 mouse melanoma cells were incubated with mouse keratinocytes or fibroblasts, followed by the isolation of melanosomes from the keratinocytes and fibroblasts. These melanosomes isolated from keratinocytes, and fibroblasts were then incubated with mouse bone marrow-derived macrophages (BMDMs) for 24 h. These macrophages or naïve macrophages were then co-grafted into C57BL/6 mice together with B16-F10 mouse melanoma cells that stably express mCherry-Luciferase (Fig. 3A). B16-F10 mouse melanoma cells co-grafted with melanosome-free macrophages (naïve) or without macrophages were used as a control. There was significantly more local melanoma growth at 21 days post injection in mice co-grafted with macrophages incubated with fibroblast-derived melanosomes compared to mice co-grafted with naïve macrophages or mice grafted with only B16-F10 cells (Fig. 3B,C). Tumors in mice co-grated with B16-F10 cells and keratinocyte-derived or melanoma-derived melanosomes did not differ from those in mice grafted with only B16-F10 cells (Fig. 3B,C). Further, analyses of mCherry signal in the tumors indicated there was a significantly higher number of melanoma cells in mice co-grafted with macrophages incubated with the fibroblast-derived melanosomes compared to the mice grafted with B16-F10 cells only, whereas co-graft with macrophages incubated with keratinocyte-derived or melanoma-derived melanosome did not result in a significant difference (Fig. 3D). This suggests that the presence of macrophages that have taken up fibroblast-derived melanosomes enhance tumor growth and proliferation.

Although the presence of macrophages incubated with melanosomes derived from melanoma cells did not differ from the control B16-F10 cell-only grafting in terms of tumor size or melanoma cell numbers (Fig. 3B–D), mice that were treated with macrophages incubated with melanosomes derived from melanoma cells had

considerably more macrophage colonization of the tumor than did mice grafted with B16-F10 cells only (Figs. 3E and EV3A). Thus, macrophages that have taken up melanoma cell-derived melanosomes enhance melanoma growth.

Next, we investigated vascularization by ultrasound imaging in the tumor region of the mice. Vascularity was similar in tumors of mice grafted with only B16-F10 cells and in those of mice co-grafted with melanoma cells and naïve macrophages or macrophages incubated with melanosomes from melanoma cells or keratinocytes (Fig. 3F; Video EV1–EV4). In contrast, there was a significant increase in vascularization in mice were co-grafted with macrophages incubated with melanosomes from fibroblasts compared to those grafted only with the melanoma cells (Fig. 3F; Video EV5). Supporting these data, we detected significantly more expression of genes, including *Cd31* and *Tie1*, in the tumors of the mice co-grafted with macrophages incubated with the fibroblast-derived melanosomes than in those only grafted with melanoma cells (Fig. 3G). CD31 and Tie1 are implicated in the development of vascular integrity and tumor angiogenesis and *Mtor*, which encodes a component of the PI3K-AKT-mTOR pathway responsible for releasing VEGF, a crucial factor in tumor angiogenesis and is also known to transform tumor-associated macrophages (Kim et al, 2013; La Porta et al, 2018; Pieniazek et al, 2018; Karar and Maity, 2011; Chen et al, 2012). These results suggest that fibroblast-derived melanosomes enhance tumor angiogenesis.

When we evaluated lymph node metastases in the mice grafted with B16-F10 cells or co-grafted with macrophages by the immunostaining with melanoma-associated antigen PNL2, lymph nodes of mice co-grafted with macrophages incubated with B16-F10-derived or keratinocyte-derived melanosomes were similar to those of controls grafted only with melanoma cells, whereas there was significantly higher PNL2 signal in mice co-grafted with macrophages incubated with fibroblast-derived melanosomes (Fig. 3H). These results suggest that the presence of macrophages that have taken up fibroblast-derived melanosomes enhances metastasis.

Next, we investigated tumor immune recognition and found significantly more infiltration of CD4$^+$ T cells, CD8$^+$ T cells, and

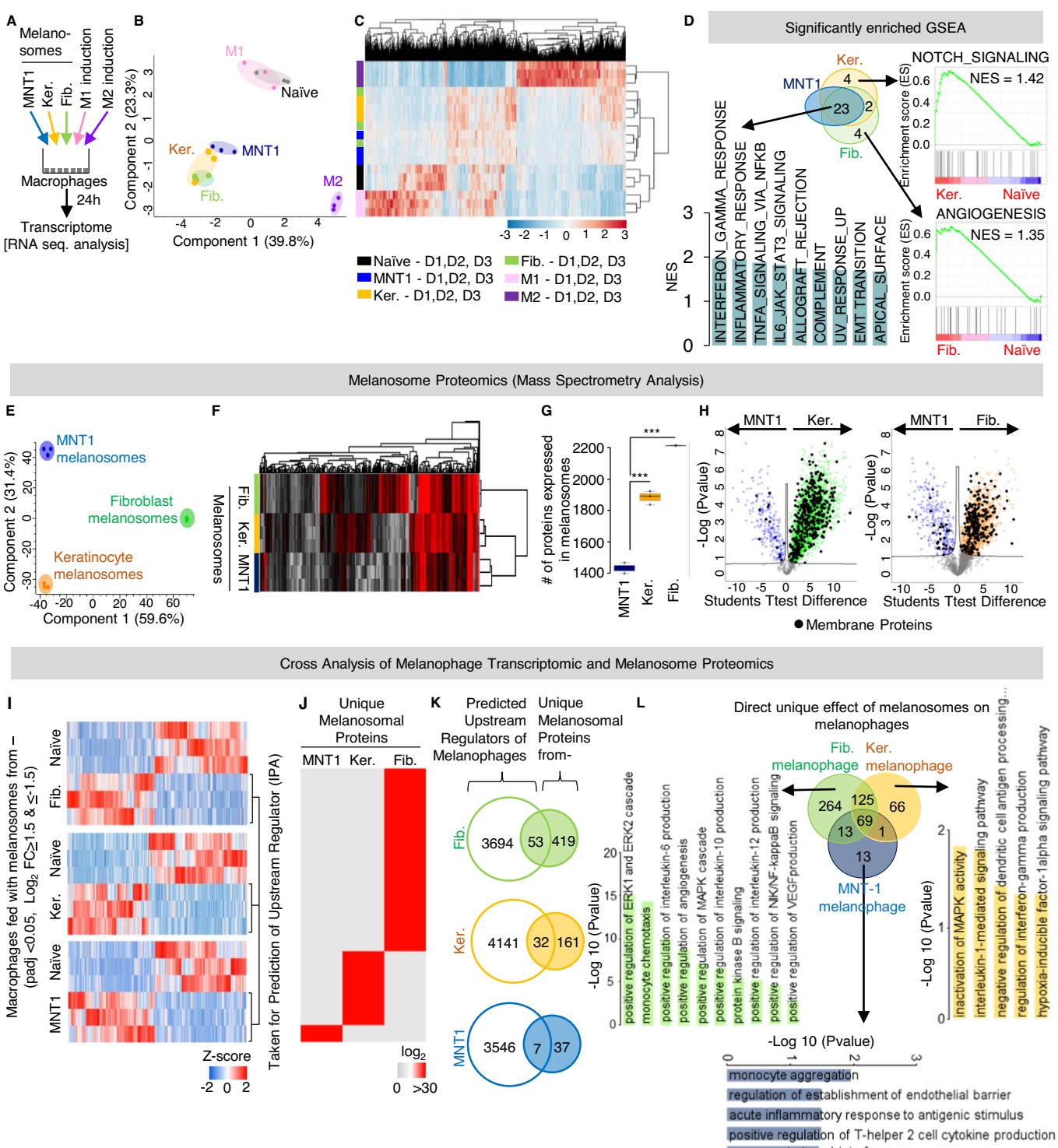

CD11b[+] myeloid cells in the tumors of the mice co-grafted with macrophages incubated with keratinocyte-derived melanosomes than in controls grafted with B16-F10 cells, whereas all other treatment groups were similar to the control (Fig. 3I–K). These results demonstrated that macrophages that have taken up keratinocyte-derived melanosomes enhance tumor immune surveillance. Subsequently, to validate the in vivo results, we

performed an experiment using the human-derived WM3682, 501mel, and MNT1 melanoma cell lines that has high proliferative potential but lacks intrinsic invasive capability, physiological characteristics that closely resemble melanoma in situ (Golan et al, 2019; Rowling et al, 2020). We found a significant increase in invasion as well as proliferation potential of melanoma cells from all lines upon treatment with conditioned media of macrophages

**Figure 4. Macrophages incubated with melanosomes isolated from different cell types are phenotypically and functionally diverse.**

(A) Workflow of transcriptomic analyses of macrophages cultured with melanosomes isolated from MNT1 cells, keratinocytes, or fibroblasts. M1-like and M2-like macrophages were used as controls. (B) Principal component analysis of transcriptomes of macrophages cultured with melanosomes isolated from MNT1 melanoma cells, keratinocytes, or fibroblasts and of naïve, M1-like, and M2-like macrophages as controls. $n = 3$ donors for each condition. (C) Heatmap of expression and clustering of transcriptomes of indicated macrophages. $n = 3$ donors for each condition. (D) Upper left: Venn diagram showing the overlap between the GSEA terms enriched in macrophages cultured with melanosomes isolated from MNT1 melanoma cells, keratinocytes, or fibroblasts. Lower left: Normalized enrichment scores for the top nine most significantly enriched Hallmark gene sets from the overlap of transcriptomes. Right: Angiogenesis Hallmark gene set unique to macrophages cultured with fibroblast melanosomes and NOTCH signaling Hallmark gene set unique to macrophages cultured with keratinocyte melanosomes compared to naïve macrophages. (E) Principal component analysis of proteomes of MNT1-, fibroblast-, and keratinocyte-derived melanosomes. $n = 3$ independent experiments. (F) Heatmap of total proteomes of indicated melanosomes showing clusters discriminating between the fibroblast-derived melanosomes (Clusters 1, 3) and keratinocyte-derived melanosomes (Cluster 2). $n = 3$ independent experiments. (G) Numbers of proteins expressed in indicated melanosomes. $n = 3$ independent experiments. (H) Volcano plots of the results of Student's $t$-tests comparing MNT1- and keratinocyte-derived melanosome proteomes (left panel) and MNT1- and fibroblast-derived melanosomes (right panel). FDR $q$ value <0.1. Blue dots are the MNT1 melanosomal proteins, orange dots are the keratinocyte melanosomal proteins, green dots are the fibroblast melanosomal proteins and black dots are the membrane proteins. An FDR $q$ value ≤0.05 was considered significant. (I) Heatmap showing the Log$_2$ fold difference in expression of genes in macrophages cultured with fibroblast-derived melanosomes (top panel), keratinocyte-derived melanosomes (middle panel), and MNT1 melanoma cell-derived melanosomes as compared to the naïve macrophages. $n = 3$ individual donors (D1, D2, and D3) for each condition. (J) Heatmap showing the LFQ expression values of proteins unique to MNT1-, keratinocyte-, and fibroblast-derived melanosomes. $n = 3$ independent experiments. (K) Overlap between the upstream regulators predicted by IPA of melanophage genes and unique melanosomal proteins. $n = 3$ individual donors for melanophages and $n = 3$ independent experiments for melanosome isolation. (L) Center: Overlap of the genes downstream to the direct effector proteins as found in (K). Graphs depict significantly enriched GO biological processes for genes unique to melanophages resulting from a culture with melanosomes from melanoma cells, fibroblasts, and keratinocytes. Data information: In panel (G), error bars represent ±SEM. An unpaired Student's $t$-test was used for statistical analysis, ***$P ≤ 0.001$ was considered significant. The centerline on the boxplot shows the medians, the box limits indicate the 25th and the 75th percentiles, and the whiskers extend to the minimum and maximum. In panel (I), an adjusted $P$ value of ≤0.05 was used as a cutoff. Log2-transformed expression of the genes were z-score normalized and visualized in hierarchical clustering with Euclidean distance method analysis (dendrograms omitted). Source data are available online for this figure.

cultured with fibroblast-derived melanosomes (isolated after treatment with melanosomes from the orthologous melanoma line or from another melanoma line). No increase in melanoma invasion potential was observed upon treatment with conditioned media of macrophages cultured with melanoma melanosomes (Figs. 3L–O and EV3B–E). When HDBECs, cells derived from the human dermal blood endothelial cells, were co-cultured with conditioned media from macrophages incubated with melanosomes derived from keratinocytes or fibroblasts there was a significant increase in tube formation compared to co-culture with conditioned media from naive macrophages (Fig. 3P). Moreover, a significant increase in the tube formation capability of HDBECs was observed upon treatment with conditioned media of macrophages that were cultured with fibroblast-derived melanosomes (either the ortholog melanoma or other melanoma line melanosomes). A smaller but significant increase in the tube formation capability of the HDBEC cells was observed upon treatment with conditioned media of macrophages cultured with melanoma melanosomes (Fig. 3Q). These findings demonstrate that macrophages that have taken up melanosomes secreted by melanoma cells promote tumor colonization, macrophages that have taken up melanosomes from fibroblasts enhance tumor growth, metastasis, and angiogenesis, and those that have taken up melanosomes from keratinocytes enhance immune surveillance.

Primary cutaneous melanoma has two distinct developmental stages, the epidermal stage, known as melanoma in situ, and the dermal stage, known as vertical melanoma; the former type of melanoma cells has proliferative abilities, and the latter invasive abilities (Golan et al, 2019). The idea that melanosomes hosted by epidermal cells and dermal cells have different effects on macrophages is, therefore, reasonable. Our data indicate that melanosomes hosted by different cells carry distinctive cues that differently polarize macrophages, which might provide a mechanistic explanation for the existing body of literature on the heterogeneous nature of macrophages in the tumor microenvironment.

## Macrophages incubated with melanosomes from different source cells are phenotypically and functionally diverse

To elucidate the fates of macrophages that have taken up melanosomes from different types of cells, we analyzed their transcriptomes (Fig. 4A). As controls, we used macrophages polarized toward the classical M1 (pro-inflammatory) or M2 (anti-inflammatory) phenotypes (Pittet et al, 2022; Ma et al, 2022). Clustering by principal component or heatmap analyses revealed a clear separation between the M1-like and M2-like macrophages, as expected (Fig. 4B,C). No overlap of the transcriptomes of macrophages that had been incubated with melanosomes from any cell source was found with either the M1 or M2-like macrophages; however, there were clear differences between melanophages resulting from treatment with melanosomes from different sources (Fig. 4B,C). This transcriptomic analysis supports the idea that tumor macrophages do not fall into the simple categories of the M1/M2 model and further supports the idea that the melanosomes from different source cells affect macrophages differently.

Analyses of differentially expressed genes revealed significant enrichment of the angiogenesis hallmark unique in macrophages incubated with fibroblast-derived melanosomes, whereas macrophages incubated with keratinocyte-derived melanosomes were significantly enriched for genes involved in NOTCH signaling (Fig. 4D). Activation of NOTCH signaling has been shown to polarize the macrophages toward a pro-inflammatory phenotype (Fukuda et al, 2012; Monsalve et al, 2009). Moreover, we found significant enrichment of pro-inflammatory IL6, TNF, and IFNγ gene sets in macrophages that had taken up melanosomes secreted by MNT1 melanoma cells and in those that had taken up melanosomes from either keratinocytes or fibroblasts (Fig. 4D). Validation of the pro-inflammatory genes including *IL6*, *TNF*, and *IL1B* revealed a significant increase in their expression in

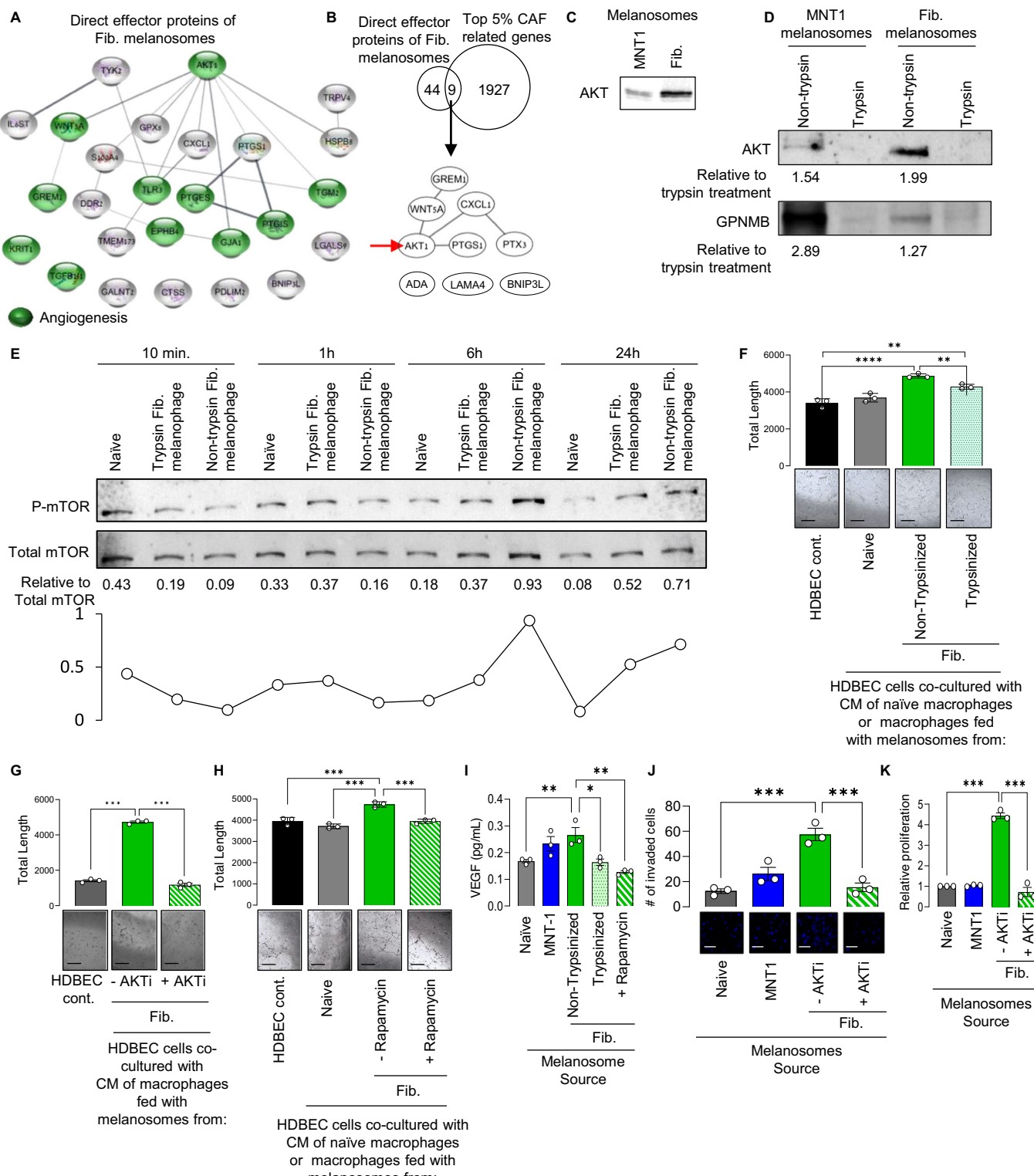

macrophages cultured with melanosomes derived from melanoma cells, keratinocytes, or fibroblasts as compared to naïve macrophages (Fig. EV4A). These results suggest that melanosomes from MNT1 melanoma cells and keratinocyte-derived melanosomes polarize macrophages toward the pro-inflammatory phenotype, whereas fibroblast-derived melanosomes result in a mixed pro- and anti-inflammatory transcriptomic phenotype. These results also confirmed that melanosomes from different host cells carry distinct cues that can polarize macrophages into diverse phenotypes.

**Figure 5. Fibroblast melanosomes contain AKT1, which induces angiogenesis via the AKT1/mTOR pathway in macrophages.**

(A) Protein–protein interaction network was identified using STRING for the direct effector proteins with positive Z-scores (Fig. EV4C). Green indicates angiogenesis-associated proteins. (B) Venn diagram of overlap between the direct fibroblast melanosomal effector proteins (Fig. 3K, found in the overlap) and proteins highly expressed in CAFs (top 5%) identified by Dror et al (2016). (C) Western blot analysis for AKT protein in MNT1- and fibroblast-derived melanosomes. (D) Western blot analysis for AKT and GPNMB protein levels in non-trypsinized or trypsinized melanosomes isolated from MNT1 cells and fibroblasts. AKT and GPNMB protein levels in each condition were normalized to the non-trypsinized melanosome protein levels. (E) Upper: Western blot analysis for phosphorylated mTOR in naïve macrophages and macrophages cultured with non-trypsinized or trypsinized melanosomes. Total mTOR was used as a loading control. Phospho-mTOR protein levels in each condition were normalized to total mTOR. Lower: Amounts of phosphorylated mTOR normalized to total mTOR. (F) Total tube lengths in cultures of HDBECs treated with conditioned media from naïve macrophages and macrophages incubated with non-trypsinized and trypsinized melanosomes. HDBEC cells and naïve macrophages were used as controls. $n = 3$ donors for conditioned media of macrophages and 3 independent angiogenesis assays. (G) Tube lengths for the HDBEC cells treated with conditioned media of naïve macrophages and macrophages cultured with fibroblast-derived melanosomes treated with vehicle or with capivasertib (AKTi). $n = 3$ donors and 3 independent experiments. Note: The representative image of the HDBEC control (cont.) has been re-used in Fig. 3Q as part of the same experiment. (H) Total tube length for the HDBEC cells treated with conditioned media of naïve macrophages and macrophages cultured with fibroblast-derived melanosomes treated with vehicle or rapamycin. HDBECs and naïve macrophages were used as controls. $n = 3$ donors and 3 independent experiments. (I) VEGF quantity in the secretome of the conditioned media of naïve macrophages and of macrophages cultured with MNT1 melanosomes or trypsinized or non-trypsinized fibroblast melanosomes and treated with vehicle or rapamycin. Monoculture of HDBECs and naïve macrophages were used as controls. $n = 3$ donors and 3 independent experiments. (J) Upper: Numbers of invaded WM3682 melanoma cells treated with conditioned media from naïve macrophages, macrophages cultured with melanoma-derived melanosomes, and macrophages cultured with fibroblast-derived melanosomes treated with vehicle or with AKTi (capivasertib). $n = 3$ independent experiments. Lower: Representative images of invasion assay. Scale bar, 50 μm. (K) Proliferation of WM3682 human melanoma cells treated with the conditioned media from naïve macrophages, macrophages cultured with melanoma-derived melanosomes, and macrophages cultured with fibroblast-derived melanosomes treated with vehicle or with AKTi. $n = 3$ independent experiments. Data information: In panels (F–K), error bars represent ±SEM. Significance was determined by one-way ANOVA. *$P \leq 0.05$, **$P \leq 0.01$, ***$P \leq 0.001$ was considered significant. Source data are available online for this figure.

We next analyzed the protein cargoes of melanosomes from different cell sources through mass spectrometry-based proteomic analysis. Most of the known melanosomal proteins (Le et al, 2021) were present in melanosomes from different cell sources (Fig. EV4B). Clustering by principal component analysis revealed a clear separation between the proteins present in MNT1 melanoma-derived melanosomes and melanosomes from the secondary cell types (Fig. 4E). Furthermore, heatmap analysis indicated two unique clusters of proteins in fibroblast-derived melanosomes and one cluster of proteins unique to keratinocyte-derived melanosomes (Fig. 4F). One cluster was shared by melanoma cell-derived melanosomes, fibroblast-derived melanosomes, and keratinocyte-derived melanosomes (Fig. 4F). This goes along with the fact the melanoma-melanosome effect on macrophage transcriptome was shared between the different melanosome sources and that fibroblast and keratinocytes-melanosome each had a unique effect on macrophage transcriptome (Fig. 4D). Melanosomes derived from the secondary cells contained significantly more protein than melanoma cell-derived melanosomes (Fig. 4G). Volcano plot analysis showed significantly more membrane-associated proteins in the fibroblast- and keratinocyte-derived melanosomes than in those derived from melanoma cells (Fig. 4H). This is in accordance with a previous report that melanosomes acquire a distinct membrane following transfer into keratinocytes (Wu and Hammer, 2014).

To investigate the drivers of diversification of the macrophages, we conducted an Ingenuity Pathway Analysis (IPA) to predict the upstream regulators of the significantly differentially expressed genes identified in the transcriptomic analysis (Fig. 4I), and we identified the proteins uniquely expressed by melanosomes from different cell sources (Fig. 4J). Next, we overlapped the predicted upstream regulator lists with the unique melanosomal protein signatures for melanosomes from each source, reasoning that this overlap should reveal melanosomal proteins that drive macrophage phenotype. There were 53, 32, and 7 proteins in the overlaps for melanosomes derived from fibroblasts, keratinocytes, and melanoma cells, respectively (Figs. 4K and EV4C–E). Annotation

analysis of these proteins showed unique enrichment of the KEGG pathways related to the positive regulation of the IL-6, IL-10, Il-12, ERK1, ERK2, and angiogenesis pathways for fibroblast-derived melanosomes (Fig. 4L), which support a mixed pro- and anti-inflammatory phenotype. In the keratinocyte- and melanoma-derived melanosome overlaps, we observed significant enrichment for pathways related to the positive regulation of the IFNγ, IL-1, T-helper 2, and type I interferons pathways indicative of pro-inflammatory signaling (Fig. 4L). Taken together, our results demonstrate that melanosomes gain a unique protein signature depending on the host cell and that the proteins in these signatures correspond to the phenotype of macrophages that have taken up these melanosomes.

## Fibroblast melanosomes loaded with AKT1 activate the AKT1/mTOR pathway to mediate angiogenesis

To identify proteins that directly polarize macrophages toward a pro-tumorigenic phenotype, we first performed hierarchical STRING analysis on the fibroblast-derived melanosome proteins since these melanosomes had the strongest pro-tumorigenic effect of those tested. These proteins were significantly enriched for KEGG pathways related to angiogenesis, with AKT1 at the top of the hierarchy (Fig. 5A). We previously showed that melanoma cell-derived melanosomes transform dermal fibroblasts into CAFs (Dror et al, 2016), and therefore we examined whether AKT1 is a CAF-related protein. Of the direct effector proteins in the fibroblast-derived melanosome signature (Fig. 4K, upper panel), nine were also among the 5% most upregulated proteins in CAFs compared to primary fibroblasts (Dror et al, 2016) (Fig. 5B). AKT1 was one of these proteins. This suggests that melanoma-derived melanosomes influence the phenotype of fibroblasts towards CAF and further generates CAF melanosomes.

Melanoma-derived small EVs can transform primary fibroblasts into CAFs (Fig. EV5A), as previously shown (Hu and Hu, 2019). Therefore, we evaluated AKT1 protein levels in fibroblast-derived melanosomes and fibroblast-derived small EVs. TEM and

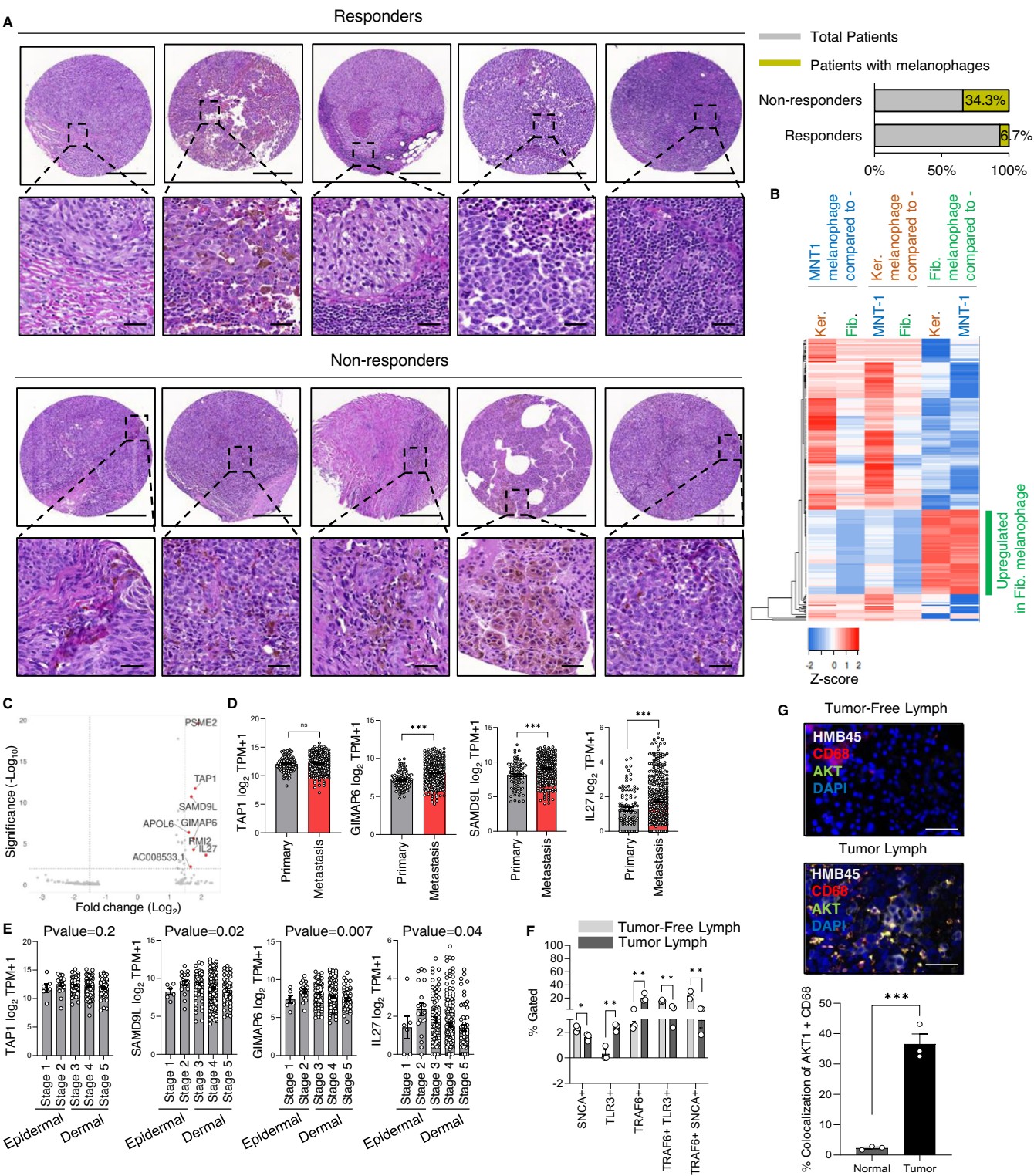

NanoSight analysis validated the presence of small EVs as derived from the fibroblast and keratinocyte cells (Fig. EV5B). AKT1 was detected only in the melanosomes (Fig. EV5C). Moreover, annotation analysis of proteins found in CAF-related melanosomes and small EVs revealed significant enrichment of proteins related to

angiogenesis in the melanosomes, whereas small EV proteins were enriched in mRNA-related processes (Fig. EV5D). Further, we found that AKT1 was present at considerably higher levels within the fibroblast-derived melanosomes cultured with MNT1, WM3682, and 501mel melanoma cells than in melanosomes from

**Figure 6. The melanosome-induced macrophage polarization signature is detected in human melanoma specimens.**

(A) Left: Representative images of H&E-stained specimens from melanoma tissue array stained with H&E of immunotherapy responders' and non-responder's cross-section. Scale bars, 400 μm. Enlarged images show the areas containing macrophages. Scale bars, 80 μm. Right: Percentages of patient samples with melanophages. $n = 15$ responders and $n = 35$ non-responders. An unpaired Student's $t$-test was used to determine significance. (B) Heatmap of $Log_2$ fold difference in gene expression ($\geq 1.2$ fold change) in macrophages incubated with fibroblast-derived melanosomes compared to $\leq 1$ fold change in MNT1 and keratinocyte melanophage. $n = 3$ donors per condition. (C) Volcano plot of top ten genes are significantly different in macrophages treated with fibroblast-derived melanosomes compared to all other treatments (naïve macrophage, MNT1, and keratinocyte melanophage). $n = 3$ donors per condition. (D) Log2 transcripts per million (TPM) $+ 1$ for *TAP1, SAMD9L, GIMAP6,* and *IL-27* in primary and metastatic lesions based on data from TCGA. Each dot represents an individual patient. $n = 104$ primary and $n = 369$ metastatic samples. (E) Log2 TPM $+ 1$ for *TAP1, SAMD9L, GIMAP6,* and *IL-27* in samples from patients in the TCGA cohort at different Clark stages of melanoma. $n = 6$ Stage1, $n = 17$ Stage 2, $n = 78$ Stage 3, $n = 168$ Stage 4, and $n = 53$ Stage 5. (F) Flow cytometry analysis showing the percentage of gated macrophage cells expressing TLR3, TRAF6, and SNCA, markers for fibroblast, keratinocyte, and melanoma-derived melanosomes, respectively, in fresh lymph node biopsies from the same patient that do and do not contain melanoma cells. $n = 1$ patient with 3 technical replicates. (G) Upper panel: Images of tumor-free and tumor-containing melanoma lymph node biopsies stained for HMB45 (white), AKT1 (green), macrophage marker CD68 (red), and DAPI for nuclei (blue). Scale bar, 20 μm. Lower panel: Percentage co-localization of AKT1 with the CD68. An unpaired Student's $t$-test was used to determine significance. Data information: In panel (D, G) an unpaired Student's $t$-test was used to determine the significance. In panel (E, F) one-way ANOVA was used to determine the significance. Error bars represent ±SEM. **$P \leq 0.01$, ***$P \leq 0.001$ was considered significant. Source data are available online for this figure.

---

melanoma cell lines (Figs. 5C and EV5E). This suggests that in CAFs, only melanosomes, and no other EVs, are loaded with the AKT1.

To determine where within melanosomes AKT1 is located, we trypsinized fibroblast- and melanoma cell-derived melanosomes to remove membrane proteins (Smith et al, 2015). AKT1 was not detected after trypsinization (Fig. 5D). As a control, we analyzed both trypsinized and non-trypsinized samples for GPNMB, a protein known to be present in membranes of melanosomes (Hoashi et al, 2010). GPNMB was only detected in the non-trypsinized samples (Fig. 5D). These results suggest that AKT1 is incorporated into the melanosome membrane in the secondary cell source.

AKT1 has been shown to induce the formation of structurally abnormal blood vessels that mimic the aberrations found in tumors through a mechanism that involves the phosphorylation of mTOR (Melick and Jewell, 2020). We, therefore, hypothesize that melanosome-localized AKT1 is a direct effector of the enhancement of angiogenesis mediated by the fibroblast-derived melanosomes. In support of this, we found that levels of phosphorylation of mTOR were significantly higher in macrophages incubated for 6 h with non-trypsinized fibroblast-derived melanosomes compared to levels in macrophages that had been incubated with trypsinized melanosomes (Fig. 5E), favoring that the fibroblast-melanosomes AKT1, mediates mTOR phosphorylation.

We next assessed the impact of fibroblast-derived melanosomes on the ability of macrophages to induce tube formation in HDBECs. We co-cultured HDBECs with conditioned media from macrophages cultured with fibroblast-derived melanosomes and observed a significant increase in tube formation compared to incubation in conditioned media from naïve macrophage culture (Fig. 5F). Melanosome trypsinization significantly reduced tube formation (Fig. 5F). Addition of an AKT inhibitor, Capivasertib (AKTi) (Andrikopoulou et al, 2022) or an mTOR inhibitor, Rapamycin (Melick and Jewell, 2020) to the macrophages that were cultured with fibroblast-derived melanosomes significantly reduced the tube formation as compared to the vehicle treatment (Fig. 5G,H).

The AKT/mTOR pathway mediates the release of VEGF (Karar and Maity, 2011), a potent angiogenic factor that is released by macrophages in response to environmental cues (Wu et al, 2010). There was significantly more VEGF in the conditioned media of macrophages incubated with the fibroblast-derived melanosomes than in the conditioned media of naïve macrophages (Fig. 5I). VEGF levels were not increased by trypsinized fibroblast-derived melanosomes or by conditioned media from macrophages incubated with fibroblast-derived melanosomes and treated with rapamycin (Fig. 5I). Further, we found an abrogation in the invasion as well as proliferation potential of the WM3682 cells treated with the conditioned media of macrophages cultured with fibroblast-derived melanosomes with the AKTi as compared to vehicle treatment (Fig. 5J,K). No increase in melanoma invasion or proliferation potential was observed upon treatment with conditioned media of macrophages cultured with melanoma melanosomes (Fig. 5J,K).

Taken together, our findings indicate that fibroblast-derived melanosomes transport AKT1 into the macrophages, which leads to mTOR phosphorylation, enhancement of VEGF secretion, and an increase in tube formation by endothelial cells and, thus, angiogenesis. However, these effects a nullified by the treating the cells with either the AKT or mTOR inhibitor.

## The melanosome-induced macrophage polarization signature is detected in human melanoma specimens

We next sought to examine whether our findings are clinically relevant. Monocytes and macrophages are present at significantly higher levels in patients who do not respond well to immunotherapy compared to patients who do (Sade-Feldman et al, 2018). Therefore, we analyzed previously described samples from patients with melanoma who did and did not respond to immunotherapy (Hoch et al, 2022) for the presence of melanophages. Based on hematoxylin and eosin (H&E) staining, melanophages were detected in 34.3% of patients in the non-responder group but only 6.7% of the subjects in the responder group (Fig. 6A). This was confirmed by Fontana Masson staining, which is specific for melanin (Appendix Fig. S2A,B). These results suggest a direct correlation between the presence of pigment-containing macrophages and treatment response.

Next, we identified the specific transcriptomic signature that defines macrophages that have taken up fibroblast-derived melanosomes. We found that 221 genes were differentially expressed in macrophages incubated with fibroblast-derived melanosomes compared to naïve macrophages (Fig. 6B). In the

top ten significantly changed genes (Fig. 6C), we identified four as effectors of melanosome function (Fig. 4K, upper panel). These four genes were *GIMAP6*, *SAMD9L*, *TAP1*, and *IL27* (Fig. 6C), which is a downstream target of AKT1 (Nadya et al, 2017). Analyses of patient data from the Cancer Genome Atlas (TCGA) showed that *GIMAP6*, *SAMD9L*, and *IL27* are significantly upregulated in metastatic tumors compared to the primary lesions, whereas *TAP1* was not differentially expressed in patient samples (Fig. 6D). Moreover, validation of those genes in the macrophages revealed their significantly higher expression, except TAP1, in the macrophages that were cultured with fibroblast-derived melanosomes (Appendix Fig. S2C), strengthening our findings of them as the direct effectors of the fibroblast melanosomes. This result suggests that the signature of macrophages that have taken up fibroblast-derived melanosomes correlates with tumor progression.

Primary tumors can be classified into five stages based on the depth of melanoma invasiveness in the epidermal and dermal regions, with the increasing stage reflecting increased tumor aggressiveness (Golan et al, 2019). There was a significant positive correlation between expression levels of *GIMAP6*, *SAMD9L*, and *IL27* with disease stage (Fig. 6E). These results suggest that dermal melanosomes are crucial in the transition of primary tumor into the dermal layer as these melanosomes polarize macrophages in a pro-tumor direction.

Next, clinically, we analyzed macrophages within fresh biopsies of lymph nodes harboring melanoma compared to tumor-free sentinel lymph node biopsies of the same patient, as macrophages are known to migrate into the sentinel lymph node (Golan et al, 2019). By flow cytometry, we analyzed the expression of proteins, including TLR3, TRAF6, and SNCA, as a marker of macrophages that have taken up Fibroblast-derived, keratinocyte-derived, and melanoma-derived melanosomes, respectively (Appendix Fig. S2D, workflow). We found that macrophages in the tumor-containing lymph nodes expressed significantly higher levels of TLR3 and TRAF6 compared to those in the tumor-free lymph nodes (Fig. 6F). Simultaneously, we found tumor macrophages expressing solely either TLR3, SNCA, or TRAF6 in the tumor-containing lymph, whereas tumor-free lymph macrophages had a significantly higher expression of the combination of these markers (Fig. 6F), suggesting the presence of the diversified tumor macrophages. Next, immunostaining for AKT1 in paired tumor-free and tumor-containing lymph node specimens showed significantly higher co-localization of AKT1 with macrophages (marked by CD68) in the tumor-containing lymph node compared to the tumor-free lymph node from the same patient (Fig. 6G). This suggests that the presence of the direct melanosomal effector protein AKT1 in macrophages is correlated with advanced disease.

## Discussion

Here we found that in the melanoma skin microenvironment, epidermal keratinocytes and dermal fibroblasts take up melanosomes secreted by melanoma cells. These melanosomes stay intact inside these cells, gain a protein signature unique to the host cell, and can be further transported into macrophages. Macrophages in the melanoma microenvironment can potentially uptake melanosomes directly from melanoma cells or via the secondary sources, epidermal keratinocytes and dermal fibroblasts. It might as well be

that macrophage incorporate melanosome in a sequential manner, by mean, first exposed to epidermal melanosome from keratinocyte, and as the tumor progress to the invasive stage (Golan et al, 2019), it acquires dermal melanosome, hosted by fibroblasts. In each stage, macrophage will be polarized a bit different with an accordance to the tumor needs, as we showed proinflammatory by keratinocyte melanosome or pro-angiogenesis by fibroblast melanosome. Taken together, we offer that melanosomes resemble the disease progression, inducing macrophage polarization toward a state that allows melanoma to progress. This process of melanosome serving as a macrophage polarization cue, might occur in normal skin upon severe wound and skin damag, since normal melanocyte melanosome are also incorporated by keratinocytes and fibroblasts in normal skin (Ando et al, 2020, 2012). In the case of damage to normal skin, melanosomes might polarize macrophages toward M1 cleaner or M2 remodeler phenotypes rather than toward a TAM phenotype, since the cargo of normal melanosomes is distinct from that of melanoma melanosomes (Dror et al, 2016).

Most studies of EVs have focused on small EVs (30–100 nm), leaving the larger EVs (200–1000 nm), like melanosomes, understudied, even though they have a significant presence in cell secretions. Small EVs are phagocytosed by the host cells and are dissolved, releasing their cargo into the host cell (Gurung et al, 2021). Our data show that it is not the case for melanosomes. These EVs stay intact, at least in some of the host cells, and can be recycled. It might be that in other physiological systems, large EVs can also be transmitted through secondary hosts. For example, similar to the pigmentation synapses (Van Den Bossche et al, 2006; Tarafder et al, 2014), larger EVs may be transmitted between cells involved in neuronal synapses (Frenguelli, 2022), immunological synapses(Dustin, 2014), and phagocytic or engulfment synapses (Li et al, 2020) might carry large EVs that are prone for recycling, harboring a message influencing physiological process. Chen et. al. showed elevated levels of PD1 in both exosomes and larger EVs obtained from metastatic melanoma patients, and that exosomes were primarily responsible for an immunosuppressive effect (Chen et al, 2018). Moreover, in another study, AKT1 and other kinases were detected in blood EVs from patients with various epithelial tumor types, which has implications for metastatic prostate cancer due to genomic abnormalities in the PI3K pathway, though the distinction between large and small EVs was not made (Minciacchi et al, 2017). It is essential to highlight that focusing exclusively on one type of EV may lead to the potential oversight of the most active subtype for the intended function.

TAMs can exert either pro- or anti-tumor effects (Ma et al, 2022; Pittet et al, 2022). The development of single-cell technologies has revealed the heterogeneity of TAMs in terms of transcript and protein expression (Ma et al, 2022) and the correlation of phenotype with various functions in tumor growth, invasion, remodeling of the extracellular matrix, vascularization, and crosstalk with other immune cells (Pittet et al, 2022). The microenvironmental cues that lead to TAMs are mostly unknown. Our findings indicate that melanosomes influence macrophage phenotype and that the cell source of the melanosomes correlates with melanoma disease stage. It is therefore possible that EVs in general and melanosomes in particular manipulate macrophage phenotypes in the tumor microenvironment.

The 501mel line has mutations in CDKN2A, BRAF, and PTEN (Tanami et al, 2004; Dahl et al, 2013); the WM3682 line has a mutation in NRAS (Lovly et al, 2012); and the MNT1 line has a mutation in BRAF (Vighi et al, 2018). Two lines have BRAF mutation, however, with additional various driver mutations, and 70–88% of primary melanomas are BRAF mutated (Castellani et al, 2023). We found that macrophages promoted melanoma aggressiveness whether they were cultured with melanosomes derived from fibroblasts that were cultured with orthologous melanosomes or melanosomes from another melanoma line. However, we cannot exclude the possibility that mutations contribute to the abilities of melanoma melanosomes to affect the fibroblasts and macrophages, either directly or via a driver mutation from the same downstream signaling (Oikonomou et al, 2014) and that the possibility that the mechanism of metastases niche formation is mutation independent.

Our findings suggest that melanophages have potential as biomarkers for prognosis and as indicators of likelihood of response to treatment. For example, our bioinformatics analysis revealed that IL-27, a downstream target of AKT, is highly correlated with the disease progression. IL-27 a known inducer of the phosphorylation of STAT3 (Hölscher et al, 2005), which regulates the expression of CD163 (Cheng et al, 2017), a classical marker of TAMs (Harris et al, 2012). Furthermore, IL-4, an anti-inflammatory macrophage stimulator, leads to the enhancement of IL-27-mediated signaling (Rückerl et al, 2006). The role of IL-27 in melanoma progression needs to be studied further. In summary, our identification of the cues that result in macrophage polarization in the melanoma microenvironment can potentially guide the development of diagnostic methods and therapies that block melanoma progression.

## Methods and protocols

### Reagents and tools table

| Reagent/resource | Reference or source | Identifier or catalog number |
|---|---|---|
| **Experimental Models** | | |
| Human whole blood | Separated from blood samples in our lab (This study) | Approval of Tel Aviv University Ethics Committee #0006436-2 |
| Human melanoma patient specimens | Tissues microarrays | BASEC:2014-0425 |
| Mouse tissue samples | Tissues obtained after sacrifice in our lab (This study) | IACUC permit #01-21-047 |
| Mouse Models:C57BL/6J | Envigo | N/A |
| **Antibodies** | | |
| Anti-CD68 (1:100 dilution) | Cell Signaling Technology | Cat #76437 S; RRID AB_2799882 |
| Anti-GPNMB (1:25 dilution for IHC and 1:100 for ICC) | R&D Systems | Cat #AF2550; RRID AB_416615 |
| Anti-HMB45 (1:50 dilution) | abcam | Cat #ab732; RRID AB_305844 |

| Reagent/resource | Reference or source | Identifier or catalog number |
|---|---|---|
| Anti-FSP1 (1:100 dilution) | abcam | Cat #ab27957; RRID AB_2183775 |
| Anti-AKT (1:100 dilution) | Cell Signaling Technology | Cat #9272; RRID AB_329827 |
| Donkey Anti-Rabbit (1:1000 dilution) | Cell Signaling Technology | Cat #8889 S; RRID AB_2716249 |
| Donkey Anti-Goat (1:1000 dilution) | Invitrogen | Cat #A11058; AB_2534105 |
| Donkey Anti-Mouse (1:1000 dilution) | Invitrogen | Cat #A31571; RRID AB_162542 |
| *Flow Cytometry* | | |
| anti-CD8a APC-Vio770 | Miltenyi Biotec | Cat #130-120-737; RRID AB_2752183 |
| anti-CD4 FITC | BioLegend | Cat #100406; RRID AB_350315 |
| anti-CD11b PE | Miltenyi Biotec | Cat #130-113-806; RRID AB_2751172 |
| anti-CD45 APC | Miltenyi Biotec | Cat #130-110-660; RRID AB_2658221 |
| anti-TRAF6 Alexa Fluor 488 | Santa Cruz Biotechnology | Cat #sc-8409; RRID: AB_628391 |
| anti-TLR3 PE | BioLegend | Cat #141903; RRID: AB_10896420 |
| anti-SNCA Alexa Fluor 594 | Santa Cruz Biotechnology | Cat # sc-12767; RRID: AB_628318 |
| CD68 APC/Cyanine 7 | BioLegend | Cat #333821; RRID: AB_2571965 |
| DAPI staining solution | Miltenyi Biotec | Cat #130-111-570 |
| *Western Blot* | | |
| DCT | Abcepta | Cat #AP8771A; RRID:AB_10553468 |
| TYR | Santa Cruz | Cat #sc-20035; RRID:AB_628420 |
| GPR143 | Bioss | Bs-11791R |
| CD9 | System Biosciences | Cat #EXOAB-CD9A-1; RRID:AB_2687469 |
| CD63 | Santa Cruz | Cat #sc-5275; RRID:AB_627877 |
| Syntenin | Santa Cruz | Cat# sc-515538 |
| anti-Mouse HRP | Cell Signalling | Cat #7076 S; RRID:AB_330924 |
| anti-Rabbit HRP | Abcam | Cat #ab6721; RRID:AB_955447 |
| Anti-Goat HRP | Abcam | Cat #ab97120; RRID:AB_10679892 |
| **Chemicals, enzymes, and other reagents** | | |
| TRIzol™ | Invitrogen | Cat #15596026 |
| Chloroform | Bio-Lab | Cat #3082301 |
| 2-Propanol | Sigma-Aldrich | Cat #278475 |
| Methanol | Bio-Lab | Cat #136806 |

| Reagent/resource | Reference or source | Identifier or catalog number |
|---|---|---|
| Paraformaldehyde, 16% | Electron Microscopy Sciences | Cat #30525-89-4 |
| Difco™ Skim Milk | Avantor Sciences | Cat #90002-594 |
| Hematoxylin Solution, Harris Modified | Sigma-Aldrich | Cat #HHS16 |
| Eosin Y solution | Sigma-Aldrich | Cat #HT110232 |
| DPX Mountant for histology | Sigma-Aldrich | Cat #06522 |
| Ketamine hydrochloride | Bremer Pharma GMBH | N/A |
| SEDAXYLAN | Eurovet animal health | N/A |
| Isoflurane, USP TerrellTM | Piramal Critical Care, Inc. | N/A |
| Biotin anti-human CD14 Antibody | BioLegend | Cat #325624 |
| MojoSort™ Streptavidin Nanobeads | BioLegend | Cat #480016 |
| PKH67 Green Fluorescent Cell Linker Kit | Sigma-Aldrich | Cat #PKH67GL-1KT |
| WST-1 reagent | Sigma-Aldrich | Cat #5015944001 |
| Synthetic melanin | MP Biomedical | Cat #8049-97-6 |
| Basement membrane extract | R & D Systems | Cat #3445-005-01 |
| Human VEGF ELISA Kit PicoKine | Boster Bio | Cat #EK0539 |
| LysoTracker | Thermo Fisher Scientific | Cat #L12492 |
| **Software** | | |
| SPSS Statistics | IBM; version 25.0 | http://www-01.ibm.com/software/uk/analytics/spss/; RRID:SCR_002865 |
| Prism | Graphpad; version 8 | http://www.graphpad.com/; RRID: SCR_002798 |
| Ingenuity Pathway Analysis | Qiagen | http://www.ingenuity.com/products/pathways_analysis.html; RRID:SCR_008653 |
| Proteome Discoverer 1.4 | Thermo Fisher Scientific | https://www.thermofisher.com/order/catalog/product/IQLAAEGABSFAKJMAUH; RRID:SCR_014477 |
| MaxQuant | Cox and Mann, 2008; version 1.5.2.8 | http://www.biochem.mpg.de/5111795/maxquant/ MaxQuant, RRID:SCR_014485 |
| Perseus software | Cox and Mann, 2008; version 1.6.15.0 | https://maxquant.net/perseus/ |
| Gene Set Enrichment Analysis | GSEA | https://www.gsea-msigdb.org/gsea/index.jsp |
| **Deposited data** | | |
| Raw mass spectrometry data Files for the melanosomes and small EVs | This study | The mass spectrometry proteomics data have been deposited to the ProteomeXchange Consortium via the PRIDE (Perez-Riverol et al, 2022) partner repository with the dataset identifier PXD046421. |
| RNA sequencing data files for macrophage cells | This study | The RNA-seq. data has been deposited in the GEO database with accession number: GSE246446. |

## Cell culture

Human MNT1, HaCat (keratinocytes), HeLa, and mouse 3T3 NIH (Fibroblasts) cells were cultured in complete DMEM (10% fetal bovine serum (FBS), 1% penicillin/streptomycin, and 1% L-glutamine; Biological Industries). MNT1 cells stably expressing RFP fused to the melanosomal protein GPR143 (Adelmann et al, 2020) were provided by Mehdi Khaled (Gustav Rousse, Paris, France) and were cultured in complete DMEM. Cells were grown at 37 °C in 5% $CO_2$ incubators.

Primary human dermal fibroblasts were isolated from neonatal foreskins by placing the foreskin in the dispase II solution (Roche) overnight. The epidermis and dermis were separated with the forceps. The dermis was digested using collagenase I solution (1 mg/mL, Sigma-Aldrich) into a single-cell suspension at 37 °C. The sample was neutralized by suspension in complete DMEM and was passed through a 70-μm strainer followed by centrifuging at $400 \times g$ for 5 min at 4 °C. Cells were cultured in complete DMEM at 37 °C in 5% $CO_2$ incubator.

Whole blood was obtained from participants recruited from Tel Aviv University (aged 25–40 years), following protocols approved by the Institutional Review Board and in accordance with the declaration of the Helsinki (#0006436-2) and upon consent from the participants. Peripheral blood mononuclear cells were isolated using Ficoll centrifugation at $1200 \times g$ for 20 min at 18 °C with brakes off. The PBMC layer was collected in a 50-mL conical tube. Monocytes were labeled with biotinylated anti-CD14 antibody (BioLegend, #325624) and isolated using the MojoSort™ Streptavidin Nanobeads (BioLegend, #480015). Monocytes were cultured in complete RPMI (10% FBS, 1% penicillin/streptomycin, and 1% L-glutamine; Biological Industries) supplemented with 5% FBS, 1% non-essential amino acids, 1% sodium pyruvate, 0.0175 mM β-mercaptoethanol, and 50 ng/mL of M-CSF (R&D Systems, #216-MC-025) to generate macrophages. Macrophages were cultured for 7 additional days with media changes every 2 days. For the differentiation into M1-like macrophages, cells were incubated for 48 h with IFNg (20 ng/mL) and LPS (75 ng/mL), and for M2-like macrophages IL13 (20 ng/mL) and IL4 (20 ng/mL) and with an additional media change for 24 h.

No cell lines used in this study appear in the database of commonly misidentified cell lines that are maintained by ICLAC and NCBI Biosample.

## Melanosome isolation

### Melanosome isolation from conditioned media

501mel, WM3682, and MNT1 melanoma cells were grown to 60% confluence, and the media was changed. On day 5, the conditioned media was collected and centrifuged at 300×g for 15 min and 1000×g for 30 min. Melanosomes were pelleted by centrifuging the supernatant at 20,000×g for 1 h as previously described (Dror et al, 2016) and were washed once by centrifuging at 5000×g for 15 min. Fibroblasts, keratinocytes, and macrophages were incubated with MNT1-derived melanosomes for 6 h. Cells were washed thoroughly twice with PBS. The conditioned media was collected from these cells on day 3, and melanosomes were isolated as described above.

### Melanosome isolation from whole cell homogenates

Keratinocytes and Fibroblasts were cultured with respective melanoma cell melanosomes for 24 h. Briefly, cells were homogenized in a glass homogenizer and the lysate was layered on the density sucrose gradient (Watabe et al, 2005). This was followed by centrifuging the gradient at 20,000×g for 1 h at 4 °C. Melanosomes were collected from the 2.0–1.8 M sucrose layer.

### Trypsinization of the melanosomes

Cleavage of surface proteins by trypsin treatment was carried out as described earlier with minor changes (Smith et al, 2015). Briefly, to remove the membrane protein segments located outside of the EVs, 1 mL of Dulbecco's phosphate-buffered saline containing 1.25% (w/v) trypsin (Biological Industries) was added to the melanosome samples, and the mixture was incubated at 37 °C for 1 h. Half of each sample was subjected to trypsin treatment, and the other half was used as a control for comparison. The effectiveness of the trypsin treatment was evaluated by performing western blotting for the membrane proteins GPNMB.

## Small EVs isolation

Small EVs were isolated by differential centrifugation as described previously with slight modifications (Lazar et al, 2015). Briefly, the conditioned media was centrifuged at 100,000×g overnight at 4 °C to deplete the media of small EVs. The MNT1 cells were grown to 70% confluence in vesicle-depleted media, and, on day 5, the conditioned media was collected and centrifuged at 300×g for 10 min, 2000×g for 10 min, and 10,000×g for 30 min. Small EVs were pelleted by centrifuging twice at 100,000×g for 1 h. Fibroblasts and keratinocytes were incubated with MNT1 small EVs for 6 h, cells were washed thoroughly twice with PBS. The media was replaced with vesicle-depleted media for an additional 3 days, and the small EVs were isolated as described above.

## Melanin quantification

Melanin content was assessed as previously described (Magina et al, 2011). Briefly, the melanosome aliquots were centrifuged at 5000×g for 20 min at 4 °C, pellets resuspended in 1 mL of 1 M NaOH in 10% DMSO, and incubated at 80 °C for 2 h. This was followed by centrifuging the aliquots at 5000×g for 20 min. The melanin concentration was determined by measuring the absorbance at 420 nm. The standard curve was generated with synthetic melanin (MP Biomedical) ranging from 0.1–1000 μg/mL. The protein concentration of the melanosome aliquots was determined using the Bradford assay.

## Hematoxylin and eosin staining

Formalin-fixed paraffin-embedded tissue sections were stained using hematoxylin (Sigma-Aldrich) and eosin (Sigma-Aldrich). Sections were mounted using the DPX mountant (Sigma-Aldrich). Images were acquired using the Aperio Slide Scanner microscope (Leica Biosystems) at 20X magnification.

## Immunofluorescence analyses

### Immunofluorescence of tissue (IHC)

Immunohistochemistry of formalin-fixed, paraffin-embedded tissues was performed as previously described (Golan et al, 2019). The compound nevi, in situ melanoma, vertical melanoma, and lymph metastasis sections were used for staining. Simultaneously we procured paired normal and metastatic lymph node sections and used them for staining. The cross-sections were stained using a macrophage marker—anti-CD68 (Cell Signaling), melanoma marker—anti-HMB45 (Abcam), melanosome marker—anti-GPNMB (R&D systems), fibroblast marker—anti-FSP1 (Abcam), pan-AKT marker (Cell Signaling) overnight at 4 °C followed by incubation in secondary antibodies Alexa Fluor 488 (Invitrogen), Alexa Fluor 594 (Cell Signaling), and Alexa Fluor 647 (Invitrogen) for 1 h at room temperature (RT). DAPI (Vector Laboratories) was used for nuclear staining. Images were obtained in an Olympus IX81 microscope and the cellSens Dimension software at 20X, 40X, and 100X magnification. Antibody information is given in the Reagents Table.

### Immunofluorescence of cells (ICC)

For immunocytochemistry, cells were fixed with 4% paraformaldehyde (PFA) for 20 min at RT. Cells were stained with primary antibody to anti-GPNMB (R&D systems), followed by incubation with Alexa Fluor 488 (Invitrogen) conjugated secondary antibodies. Cells were stained for nuclear staining with DAPI (vector laboratories) and imaged using Olympus IX81 microscope and the cellSens Dimension software at 10 μm magnification. Antibody information is given Reagents Table.

## Transmission electron microscopy

Melanosomes were fixed using Karnovsky's fixative and processed as described previously (Dror et al, 2016). Samples were dehydrated in a series of ethanol concentrations and embedded in araldite, followed by sectioning with a diamond knife on an LKB 3 microtome into an ultrathin section of 80 nm. Sections were collected onto 200-mesh, thin bar copper grids. The sections were stained using uranyl acetate and lead citrate for 10 min each and viewed under Tecnai TEM 100 kV (Phillips) equipped with a MegaView II CCD (charge-coupled device) camera and analysis version 3.0 software.

For negative staining, the particles were adsorbed from suspension onto a carbon-coated formvar grid and were embedded in a thin film of uranyl acetate-methylcellulose (0.5% uranyl acetate (Merck) in 0.5% methylcellulose, 25 centipoise (Sigma). The ultrathin sections were analyzed by sequentially fixing the pellets

4% formaldehyde/1% glutaraldehyde, osmium, and ethanolic uranyl, followed by embedding in epoxide (Sigma-Aldrich) according to the manufacturer's instructions. The ultrathin sections of 50 nm were post-stained using uranyl and lead. Images were acquired using a ZEISS EM 912 transmission electron microscope equipped with a Proscan CCD camera (TRS).

The EVs were adsorbed to Carbon-Formvar-coated copper grids. Grids were stained with 1% (w/v) uranyl acetate and air-dried. Samples were viewed with Tecnai 12 TEM 120 kV (Phillips) equipped with Phurona camera and RADIUS software (Emsis GmbH).

## NanoSight analysis

Melanosomes or small EVs were quantified using the nanoparticle tracking analysis on a NanoSight LM10 equipped with a 405 nm laser (Malvern Instruments). Data were analyzed with the NTA 3.0 software. The melanosome stock was diluted to 10 μg/mL protein concentration, and each sample was analyzed three times for 60 s. The mean values were used.

## Indirect co-culture assay

MNT1 cells stably expressing RFP fused to GPR143 were seeded into 12-well hanging inserts with a pore size of 0.45 μm at $8 \times 10^4$ seeding density. Fibroblasts, keratinocytes, or macrophages were seeded in the 12-well cell culture plates. On the day of the co-culture, the hanging inserts were moved on top of the 12-well cell culture plates, and samples were co-cultured for 24 h and imaged. Next, fibroblasts or keratinocytes expressing RFP were seeded onto the hanging inserts and were indirectly co-cultured further with naïve macrophages for 24 h. Cells were fixed using 4% PFA for 20 min at room temperature and stained with DAPI (Vector Laboratories). Cells were imaged using Olympus IX81 microscope and the cellSens Dimension software at 20 μm magnification. Macrophages were fixed and imaged as described above.

## Conditioned media preparation

Macrophages were incubated with melanosomes secreted by MNT1 cells or melanosomes isolated from keratinocytes or fibroblasts. Naïve macrophages were used as controls. The supernatant was collected on ice and centrifuged at 1000×*g* for 5 min to remove cellular debris and cells. The conditioned media was aliquoted and stored at −80 °C until use.

## Pkh67 labeling

Melanosomes and small EVs were pelleted as described above. The pellets were resuspended in 1 mL of Diluent C and incubated for 5 min at room temperature with the Pkh67 dye (Sigma-Aldrich). The dye was quenched using 10% bovine serum albumin (BSA), and the labeled vesicles were centrifuged at 100,000×*g* for 70 min at 4 °C on a 0.961 M sucrose density gradient. The pellet was washed once with PBS, and melanosomes and small EVs were obtained by centrifuging at 5000×*g* and 100,000×*g*, respectively, for 20 min at 4 °C.

## Invasion assay

Prior to the invasion assay, WM3682 melanoma cells were co-cultured for 24 h with the conditioned media from macrophages that had been incubated with melanosomes secreted by MNT1 cells or melanosomes isolated from keratinocytes or fibroblasts. Naïve macrophage conditioned media was used as a control. Melanoma cells were trypsinized and seeded at $2 \times 10^4$ cells/insert in serum-free DMEM. The pore of the transwell membrane insert was 8 μm and was coated with 0.1% BSA for 1 h, followed by coating with Matrigel (BD Biosciences) for 1 h. The cells were allowed to invade for 9 h after the addition of the complete DMEM in the lower chamber of the transwell system. The cells were fixed with ice-cold methanol for 20 min at room temperature, followed by DAPI staining. Images were acquired using the Olympus IX81 microscope and the cellSens Dimension software. Three fields per insert membrane were photographed, and the number of invaded cells were determined.

## Proliferation assay

WM3682 melanoma cells were seeded in a 96-well plate at a seeding density of $1 \times 10^4$ cells/well and cultured for 24 h with the conditioned media from macrophages that had been incubated with melanosomes secreted by MNT1 cells or melanosomes isolated from keratinocytes or fibroblasts. Naïve macrophage conditioned media was used as control. WST-1 reagent (Sigma-Aldrich) was used to quantify proliferation according to the manufacturer's instructions. The absorbance was measured at 440 nm using an ELISA plate reader.

## LysoTracker assay

Melanosomes derived from MNT1, keratinocytes, or fibroblast cells were labeled using the PKH67 dye. Macrophages were than incubated with respective melanosomes from different sources for 24 h followed by trypsinization and centrifuging the cell pellet. This was followed by staining with the 70 nm LysoTracker dye in cell culture media for 30 min at 37 °C and washed with PBS X1. The cells were imaged/quantified using the ImageStream cytometry software.

## Tube formation assay

Prior to the assay, human dermal blood endothelial cells (HDBECs) were starved overnight in RPMI media or with the EBM media containing 2.5% FBS. For the tube formation assay, 48-well cell culture plates were coated with 10 μg/mL of Cultrex Basement Membrane Extract (R&D Systems, #3445-005-01) and incubated for 30 min at 37 °C before cell seeding. This was followed by trypsinization of the cells and plating on the Basement Membrane Extract at a seeding density of $2.5 \times 10^3$ cells/well and incubated for 4 h. Tube formation analysis was performed using ImageJ software Angiogenesis analyzer. Images were acquired using a Nikon Eclipse TS-100 microscope at 10X objective and a Nikon DS-Fi1 camera. Experimenter were blinded towards the analysis for this experiment.

## Western blot assay

Melanosome aliquots were centrifuged at 5000×*g* for 20 min and the pellet was lysed using RIPA lysis buffer with inhibitors and incubated on ice for 2 h followed by centrifuging at 12,000×*g* for 20 min at 4 °C. The clear phase was stored at −80 °C until further use.

An equal amount of proteins as determined by Bradford (80 µg) were resolved on 10% (for mTOR, CD9, CD63, Syntenin, GPNMB, GPR143, DCT, and TYR) and 8% (for AKT and p-mTOR) SDS-polyacrylamide gels, followed by electro-blotting onto a nitrocellulose membrane (GE Healthcare). The membrane was stained with anti-TYR (Santa Cruz), anti-DCT (Abcepta), anti-GPR143 (Bioss), anti-GPNMB (R&D Systems), anti-CD9 (System Biosciences), anti-CD63 (Santa Cruz), anti-Syntenin (Santa Cruz), anti-AKT (Cell Signaling), anti-mTOR (Abcam), and anti-phospho-mTOR (Abcam) primary antibodies overnight at 4 °C. Proteins were visualized using SuperSignal Chemiluminescent Substrates (Pierce) using horseradish peroxidase-conjugated secondary antibodies to anti-rabbit IgG (Cell Signaling), anti-mouse IgG (Cell Signaling), and anti-goat IgG (Abcam), respectively. Additional information on antibodies is given in the Reagents Table. Raw images for the western blots are given in Appendix Figs. S3–S5.

## VEGF ELISA

Conditioned media from macrophages incubated with melanosomes secreted by MNT1 cells or melanosomes isolated from keratinocytes or fibroblasts and conditioned media of macrophages that had been incubated with trypsinized melanosomes from fibroblasts or with non-trypsinized fibroblast melanosomes treated with 100 nM rapamycin were analyzed. The undiluted conditioned media was tested using the VEGFA ELISA kit (Boster Bio) according to the manufacturer's instructions.

## Mouse melanoma model

### Cell culture
B16-F10 melanoma cells and 3T3 NIH cells were cultured in a complete DMEM. B16-F10 mCherry cells expressing firefly luciferase reporter were cultured in a complete DMEM containing puromycin. Primary mouse keratinocytes were isolated from the dorsal skin of newborn mice. Briefly, the dorsal skin of newborn mice was extracted and exposed to dispase (Sigma-Aldrich) to isolate the epidermis followed by the trypsin (Biological Industries) treatment of the epidermis. The resulting keratinocytes were first cultivated on fibroblast feeder cells for four cycles and then moved to tissue culture dishes without feeder cells.

### Bone marrow-derived macrophages
For the generation of BMDMs, 8-week-old mice were used. The bone marrow cells were isolated from the femur and tibia bones of the mice. Cells were seeded in 100-mm tissue culture plates with a complete DMEM. The following day the non-adherent cells were collected and seeded in complete DMEM supplemented with 100 ng/mL of M-CSF. BMDMs were cultured for 7 additional days with fresh media change every 2 days.

### Mouse melanoma model
Female C57BL/6 mice, aged 7–8 weeks, were procured from Envigo and allowed a period of 1 week for acclimatization after arrival. The animals were maintained under pathogen-free conditions with a controlled light/dark cycle of 12 h each, at a regulated temperature of 22 ± 1 °C and relative humidity of 32–35%. They were provided with ad libitum access to food and water. All the animal experiments were conducted according to the guidelines of the Tel Aviv University Institutional Animal Care and Use Committee (#01-21-047). The mice were randomly assigned into five groups. Investigators were not blinded to allocation during the experiments and outcome assessment, except for the ultrasound experiment, which was performed blinded by an external member to determine the vascularization. B16-F10 mCherry cells expressing firefly luciferase reporter ($1 \times 10^5$ cells) were injected alone or co-grafted with naïve BMDMs or BMDMs ($5 \times 10^5$ cells) [ratio of 1:5, (Lerner et al, 2023)] incubated with melanosomes secreted by B16-F10 cells or isolated from keratinocytes or 3T3 NIH fibroblasts in 0.1 mL sterile PBS. The cells were injected subcutaneously into the dorsal side of the C57BL/6 mice as previously described (Dror et al, 2016). At 7-, 14-, and 21-days post injection (DPI), the tumor size was determined using the bioluminescence imaging after injecting the animals with 100 µL of luciferin (Promega) using the IVIS spectrum system (Caliper Life Sciences, PerkinElmer) and with a tumor caliper. A total of 32 mice were used: $n = 5$ B16-mCherry control, $n = 5$ B16-mCherry co-inoculated with naïve BMDM, $n = 8$ B16-mCherry co-inoculated with BMDMs incubated with B16-F10 melanosomes, $n = 6$ B16-mCherry co-inoculated with BMDMs incubated with melanosomes isolated from keratinocytes, and $n = 8$ B16-mCherry co-inoculated with BMDMs incubated with melanosomes isolated from 3T3 NIH fibroblasts. At 7-, 14-, and 21-days post injection (DPI), animals were injected with 100 µL of luciferin (15 mg/mL, Promega), and the tumor size was determined using the IVIS spectrum system (Caliper Life Sciences, PerkinElmer). At 21 DPI, the mice were sacrificed, and tumors and inguinal lymph nodes were surgically resected.

About $1 \times 10^5$ cells B16-F10 mCherry cells expressing firefly luciferase reporter were injected to avoid a dominant effect of melanoma cell growth that might overshadow the subtle effects of the macrophages. Further, formulation of injection is critical as this provides a surface for the cancer cells to grow and enhances engraftment at the tumor inoculation site (Quintana et al, 2008). Specifically, angiogenesis is affected in the presence of Matrigel (Kitahara et al, 2010). Therefore, to avoid this bias in favor of melanoma cell growth, we formulated the cells in PBS. We used a ratio of melanoma cells to tumor environment cells of 1:5, as was done previously (Lerner et al, 2023), in order to allow the observation of the environmental cell effect without any bias toward melanoma growth.

### Tumor volume measurement
At 7, 14, and 21 DPI, tumor volumes were estimated based on caliper measurements using the formula: volume = 0.52 $ab^2$, where a = long diameter and b = short diameter.

### Ultrasound imaging
Anesthetized mice were placed on a heating pad to maintain body temperature and to minimize temperature-induced changes in blood flow. The hair surrounding the imaged area was removed

before imaging. Ultrasound imaging was performed using a Vevo® 3100 high-frequency imaging system (FUJIFILM VisualSonics Inc.) equipped with an MX550D transducer (40 μm axial resolution). B-mode images were taken at a center frequency of 40 MHz, followed by Color Doppler Imaging for vascularization.

### Colonization assessment

Colonization assessment was independent of the size or spread of the tumor and was based on the visual counting of the development of tumor in a mouse.

### Isolation of primary cells from tumors

Tumors were digested using collagenase IV (2 mg/mL) and DNase I (2000 U/mL) (both from Sigma-Aldrich) in RPMI media with 0% FBS and 1% penicillin/1% streptomycin for 20 min at 37 °C. Cells were then passed through a 70-μm strainer (Thermo Fisher) and centrifuged at 700×g for 5 min at 4 °C. The single-cell suspensions were analyzed by flow cytometry.

### Flow cytometry

For cell-surface staining, monoclonal antibodies conjugated to APC-Vio® 770, FITC, PE, and APC and specific for the following antigens were used: CD8α, CD4, CD11b, and CD45, all from Miltenyi Biotech. Antibody labeling was performed in PBS containing 1% FBS for 20 min at 4 °C. Flow cytometry analysis was performed on Cytoflex4L (Beckman Coulter), and datasets were analyzed using Kaluza software. DAPI (Miltenyi Biotech) was used to determine the live/dead cell population.

### qRT-PCR and RNA from formalin-fixed, paraffin-embedded (FFPE) tissues

Section of 30 μm of formalin-fixed, paraffin-embedded tumors were subjected to de-paraffinization with 100% xylene for 3 min at 50 °C, followed by centrifugation at 14,000×g for 1 min. The pellet was washed twice with absolute ethanol and air-dried. To digest proteins, 150 μl of 1X protease K digestion buffer containing 20 mM Tris-HCl, pH 8, 1 mM CaCl₂, 0.5% sodium dodecyl sulfate, and 500 μg/ml proteinase K (New England Biolabs) was added to the pellet, which was then incubated at 55 °C for 3 h. RNA was isolated using the TRIzol™ protocol (Invitrogen) according to the manufacturer's instructions and stored at −80 °C until further processing. For mRNA analysis, cDNA was generated using the qScript cDNA synthesis kit (Quantabio) and subjected to qRT-PCR using PerfeCTa SYBR green FastMix (Quantabio). The resulting data were expressed as fold changes relative to the naïve BMDM control or B16-F10 cell-only control. The experiments were repeated in three randomly selected mice per condition to ensure reproducibility. The primer sequences utilized for the analysis are listed in Appendix Table S1.

## Proteolysis and mass spectrometry analysis

Proteins from the melanosome samples (three independent experiments for each condition) were precipitated with 90% ethanol at 90 °C followed by centrifugation at 11,200×g for 5 min. The supernatant was dried and resuspended in 9 M urea, 400 mM ammonium bicarbonate, and 3 mM DTT. After 30 min at 60 °C, the sample were reduced by incubation in 12 mM iodoacetamide in 200 mM ammonium bicarbonate in the dark at room temperature for 30 min. Finally, samples were digested with modified trypsin (Promega) at a 1:50 enzyme-substrate ratio in 1 M urea, 50 mM ammonium bicarbonate for 2 h at 37 °C. Tryptic peptides were desalted using the C18 tips (Top Tip, Glygen), dried, and resuspended in 0.1% formic acid. Peptides were resolved by reverse-phase chromatography on 0.075 mm × 180 mm fused silica capillaries (J&W Pharmalab) packed with Reprosis reverse-phase material (Dr. Maisch GmbH). Peptides were eluted with a 60-min linear gradient from 5 to 28%, then a 15-min linear gradient from 28 to 95%, 25 min at 95% acetonitrile with 0.1% formic acid in water, at a flow rate of 0.15 μl/min. Mass spectrometry was performed with a Q Exactive HF mass spectrometer (Thermo Fisher Scientific) in a positive mode with a full MS scan followed by collision-induced dissociation of the 18 most dominant ions selected from the first MS scan. The mass spectra were analyzed using the MaxQuant software 1.5.2.8 (Cox and Mann, 2008), and the data were quantified by label-free analysis using the same software. Statistical analysis was done using Perseus 1.6.15.0 software (Cox and Mann, 2008).

The heatmap and the principal component analysis shown in Fig. 3F,H were performed using the Perseus software (1.6.15.0) by taking the whole proteome into consideration for the analysis. Volcano plot analysis for Fig. 3H was performed using the Perseus 1.6.15.0 software. Heatmap for Fig. 3K was generated using the excel file by marking unique proteins expressed by melanosomes from various cell source.

The mass spectrometry proteomics data have been deposited to the ProteomeXchange Consortium via the PRIDE (Perez-Riverol et al, 2022) partner repository with the dataset identifier PXD046421.

The mass spectrometry data were available via the PRIDE database with accession number PXD046421.

## RNA sequencing and data analysis

RNA was isolated using the TRIzol™ protocol (Invitrogen) according to the manufacturer's instructions and stored at −80 °C until further processing. Libraries were prepared from ~50 ng of total RNA using the NEBNext Ultra II RNA Library Prep kit and NEBNext Poly(A) mRNA Magnetic Isolation Module, according to the manufacturer's protocol (NEB). Libraries' amplification was performed with 15 PCR cycles. Quality assessment for all libraries was performed using TapeStation 4200 system (Agilent) and Qubit 3.0 Fluorometer (Invitrogen). Sequencing was performed using a NextSeq 500/550 High Output Kit v2.5 (75 Cycles) kit on a NextSeq 500 system, according to the manufacturer's instructions (Illumina). Library preparation and sequencing were performed at the Genomics Research Unit, Life Sciences Inter-Departmental Research Facility Unit, Tel Aviv.

Raw reads were processed with cutadapt, v3.4, to remove low-quality and adapter sequences, then aligned to the human GRCh38 genome with TopHat, v2.1.1, allowing for 5 mismatches. Gene annotations from Ensembl release 99 were used to assign raw counts per gene per sample with htseq-count, v0.13.5. Normalization and differential expression analysis were done with the DESeq2 package, v1.30.0, after discarding genes with less than 10 reads in the whole system. The statistical model included the exposure to melanosomes as well as the donor factor; thus comparisons were calculated taking the

donor into account. Plots, such as heatmaps, were produced with normalized counts after batch correction with respect to the donor. Batch correction was done with the ComBat function of the sva package, v3.38.0. Pair-wise comparisons were tested with default parameters, except that the independent filtering algorithm was not used. The significance threshold was taken as $p$adj < 0.05. Genes with a significant change in expression were further filtered according to the extent of the change, in a baseMean-dependent manner. The requirement was for an absolute log2FoldChange higher than 5/sqrt(baseMean) + 0.4.

Gene set enrichment analysis (GSEA) was performed on the significantly ($P$ adjusted value ≤ 0.05) differentially expressed genes for each condition compared to the naïve macrophages. A fold change value of greater than 1.5 and less than −1.5 was taken into consideration for the ranking which was done using the R programming software. The GSEA was done using the GSEA v3.0 from the Broad Institute using the Hallmark gene sets. The top 25 significantly enriched gene sets were taken into consideration.

The IPA core analysis (QIAGEN) was applied to the gene symbols and significantly differentially expressed genes with an absolute log2-transformed fold change greater than 1.5 or less than −1.5. This was used to generate potential upstream regulators by matching with the Human Ingenuity Knowledge Database, which provided $P$ value and activation Z-score values to infer the activation state (increase or decrease). The likelihood of over-representation in the dataset for upstream analysis was determined using Fisher's right-tailed exact test.

The GO annotation analysis was performed using the DAVID software on the genes downstream to the direct melanosomal effector proteins found in the overlap of Fig. 3L for each melanophages. The downstream genes were predicted by the IPA software.

For the unique transcriptomic signature gene analysis, log₂ expression of genes highly represented (i.e., with ≥1.2 fold change value in macrophages cultured with fibroblast melanosomes compared to ≤1 fold change expression values in macrophages cultured with melanosomes from keratinocytes and MNT1 cells). Genes that were differentially expressed in macrophages treated with fibroblast-derived melanosomes compared to naïve macrophages were used for the further filtering of the transcriptomic gene signature. The volcano plot was generated using the https://huygens.science.uva.nl/VolcaNoseR2/ web tool (Goedhart and Luijsterburg, 2020).

The RNA-seq. dataset is available in the following Gene Expression Omnibus GSE246446.

### The Cancer Genome Atlas dataset

Bulk RNA-seq. and clinical information of the patients from the skin cutaneous melanoma were downloaded from the UCSC XENA web browser http://xena.ucsc.edu/. The Log₂ transcripts per million (TPM) + 1 values were extracted for *GIMAP6*, *IL-27*, *TAP1*, and *SAMD9L* genes.

### Tissue microarray

A TMA was prepared from FFPE samples of 69 patients with stage III and IV metastatic melanoma who were treated at the University Hospital of Zürich under ethics approval BASEC:2014-0425. Only

patients with the diagnosis of responders (R) or non-responders (NR) after 3 months and 12 months of immunotherapy were taken into consideration.

### Melanoma sample processing

Fresh biopsies of lymph nodes harboring melanoma or tumor-free lymph nodes from the same patient were acquired from the Tel Aviv Sourasky Medical Center (Helsinki# 0016-20-TLV). Biopsies were washed with PBS X1 and separated into single cells. Briefly, biopsies were minced and incubated at 37 °C for 25 min in serum-free DMEM containing collagenase IV (Worthington) and deoxyribonuclease I (Worthington). DMEM supplemented with 10% FBS was used to stop the enzymatic reaction. The cell suspension was filtered through a 70-µm cell strainer (Corning). The cells were then centrifuged for 5 min at 500×g at 10 °C. The pellet was reconstituted in Red Cell Lysis Buffer (Sigma-Aldrich) according to the manufacturer's protocol. The pellet were subjected to flow cytometry staining with the antibodies against APC, Alexa Fluor 488, PE, Alexa Fluor 594, and APC/Cyanine 7, specifically for the following antigens: CD45 (Miltenyi Biotech), TRAF6 (Santa Cruz), TLR3 (BioLegend), SNCA (Santa Cruz), and CD68 (BioLegend) respectively. Antibody labeling was performed in PBS containing 1% FBS for 20 min at 4 °C. Flow cytometry analysis was performed on Cytoflex5L (Beckman Coulter), and datasets were analyzed using Kaluza software. Additional information on antibodies is given in the Reagents Table.

### Fontana Masson staining

The tissue microarray was subjected to melanin staining using the Fontana Masson method as recommended by the manufacturer (Abcam, ab150669). Images were captured using the Aperio Slide Scanner at a magnification of 40X.

### Statistics

Statistical analyses were performed in GraphPad Prism Software version 9.0. For analysis of more than two groups, one-way ANOVA was performed with Tukey's correction performed for multiple comparison analyses. We utilized Student's unpaired $t$-test with Welch's correction for comparison between the two groups. For the volcano plot and the GSEA analysis, the FDR $q$ value of ≤0.05 was taken into consideration. To evaluate the differences among stages of melanoma, we used a test for the linear trend as a follow-up multiple testing after one-way ANOVA.

## Data availability

The datasets produced in this study are available in the following repositories: RNA-Seq data: Gene Expression Omnibus GSE246446. Mass Spectrometry data: PRoteomics IDEntifications [PRIDE] PXD046421.

The source data of this paper are collected in the following database record: biostudies:S-SCDT-10_1038-S44318-024-00103-7.

## Peer review information

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

## Acknowledgements

This work was supported in part by the European Research Council (ERC) under the European Union's Horizon 2020 research and innovation program (grant agreement no. 726225), the I-CORE Gene Regulation in Complex Human Disease Center (no. 41/11), the Melanoma Research Alliance (MRA; grant 402792), and Israel Science Foundation (grant 129/13). We thank the Smoler Proteomics Center, Faculty of Biology of the Technion Institute of Technology, for performing mass spectrometry and the Genomics Research Unit (GRU), Faculty of Life Sciences, Tel Aviv University, for performing the RNA sequencing.

## Author contributions

**Roma Parikh**: Conceptualization; Data curation; Software; Formal analysis; Validation; Investigation; Visualization; Methodology; Writing—original draft; Writing—review and editing. **Shivang Parikh**: Data curation; Validation; Methodology. **Daniella Berzin**: Data curation; Validation; Methodology. **Hananya Vaknine**: Formal analysis; Investigation. **Shai Ovadia**: Data curation; Software; Formal analysis. **Daniela Likonen**: Data curation; Visualization; Methodology. **Shoshana Greenberger**: Data curation; Validation; Methodology. **Alon Scope**: Visualization; Methodology. **Sharona Elgavish**: Data curation; Software; Formal analysis. **Yuval Nevo**: Data curation; Software; Formal analysis. **Inbar Plaschkes**: Data curation; Software; Formal analysis. **Eran Nizri**: Resources. **Oren Kobiler**: Data curation. **Avishai Maliah**: Data curation. **Laureen Zaremba**: Data curation; Software; Formal analysis. **Vishnu Mohan**: Data curation; Validation; Methodology. **Irit Sagi**: Visualization; Methodology. **Ruth Ashery-Padan**: Visualization; Methodology. **Yaron Carmi**: Formal analysis; Investigation. **Chen Luxenburg**: Formal analysis; Investigation. **Jörg, D. Hoheisel**: Data curation; Software; Formal analysis. **Mehdi Khaled**: Visualization; Methodology. **Mitchell Levesque**: Resources. **Carmit Levy**: Conceptualization; Resources; Data curation; Software; Formal analysis; Supervision; Funding acquisition; Validation; Investigation; Visualization; Methodology; Writing—original draft; Project administration; Writing—review and editing.

Source data underlying figure panels in this paper may have individual authorship assigned. Where available, figure panel/source data authorship is listed in the following database record: biostudies:S-SCDT-10_1038-S44318-024-00103-7.

## Disclosure and competing interests statement

The authors declare no competing interests.

# Expanded View Figures

**Figure EV1. Associated with Fig. 1: Melanosomes are detected in non-cancerous cells in the tumor microenvironment.**

(A) Images of compound nevi, in situ and vertical melanoma and lymph metastasis samples from two patients. Upper panels: Images of H&E-stained sections of compound nevi, in situ, and vertical melanoma and lymph metastasis samples from two patients. Black dashed lines demarcate the epidermal and dermal borders. Scale bars, 200 μm. Inset images show macrophages containing pigmented vesicles. Scale bars, 50 μm. Lower panels: Images of consecutive specimens stained for HMB45 (white), melanosomal marker GPNMB (green), and macrophage marker CD68 (red) and with DAPI for nuclei (blue). White dashed lines demarcate the epidermal and dermal borders. Scale bars, 200 μm. Inset images show the co-localization of macrophages with melanosomes. Scale bars, 50 μm. (B) Left: Images of H&E-stained sections of in situ melanoma specimens from two patients. Black dashed lines demarcate the epidermal and dermal borders. Scale bars, 200 μm. Right: Immunofluorescence images of consecutive samples stained for CD68 (red) and GPNMB (green) or FSP1 (red) and GPNMB (green). White dashed lines demarcate the epidermal and dermal borders. Scale bars, 200 μm. Inset images show keratinocytes within the epidermis containing melanosomes marked with white arrowheads. Scale bars, 50 μm.

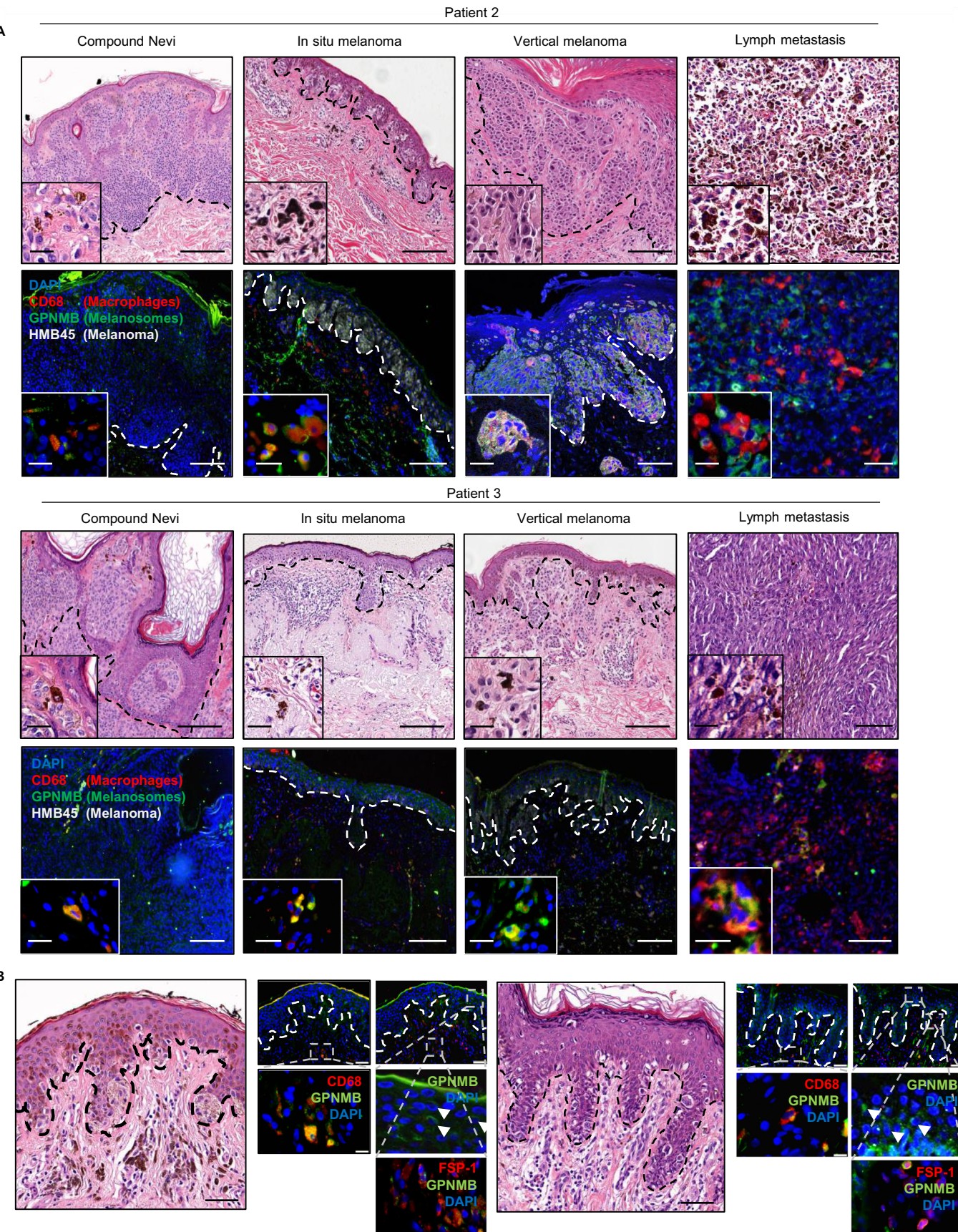

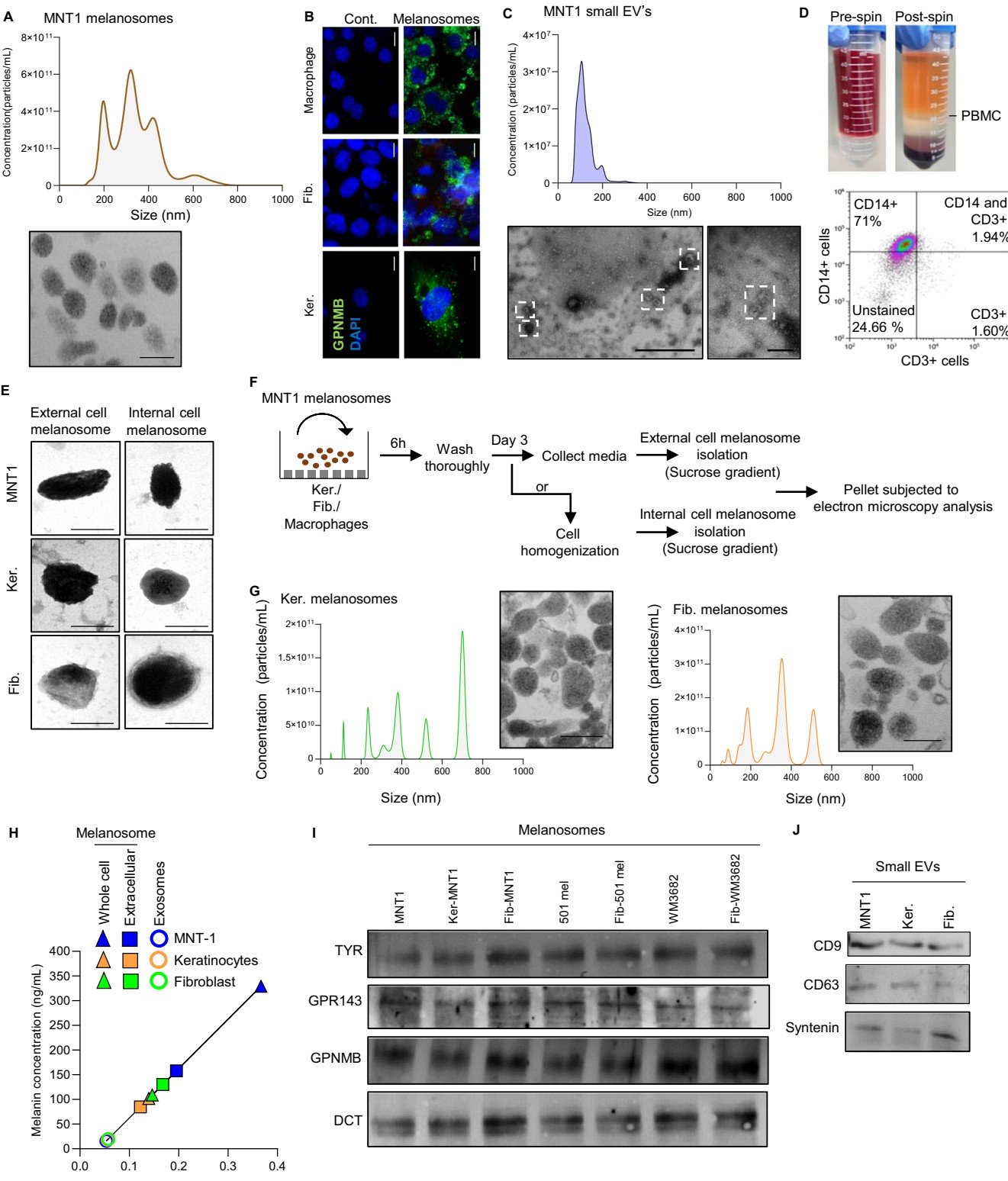

◀ **Figure EV2.   Associated with Fig. 2: Cell-to-cell transfer of melanoma cell-derived melanosomes occurs in the tumor microenvironment.**

(A) NanoSight analysis (upper) and TEM analysis (lower) of melanosomes secreted from melanoma cells. Scale bars, 0.5 μm. (B) Immunofluorescence images of control keratinocytes, fibroblasts, and naïve macrophages and of cells cultured with melanosomes for 24 h. DAPI-stained nuclei appear blue. Scale bars, 10 μm. (C) NanoSight analysis (upper) and TEM analysis (lower) of small EVs secreted from melanoma cells. Scale bars, 0.5 μm left image and 0.2 μm for right image. Small EVs are highlighted with white dashed boxes. (D) Top: Photograph of peripheral blood mononuclear cell sample after Ficoll gradient centrifugation. Bottom: Flow cytometry analysis of monocytes isolated using a CD14$^+$ cell isolation kit; 71% of isolated cells were CD14$^+$. (E) TEM images of melanosomes from inside cells (right) and from conditioned media (left) of indicated samples. Scale bars, 0.2 μm. (F) Workflow of isolation of melanosomes from inside cells and from conditioned media after 3 days of culture. (G) NanoSight analysis (right) and TEM images (left) of melanosomes isolated from the keratinocyte and fibroblast cells cultured with MNT1 melanoma cell melanosomes. Scale bars, 0.5 μm. (H) Quantification of melanin concentration in samples isolated by differential centrifugation, gradient centrifugation, and ultra-centrifugation for secreted melanosomes, internalized melanosomes, and small EVs, respectively, from melanoma cells, keratinocytes, and fibroblasts; $n = 2$ independent experiments. (I) Western blot analysis for signature melanosomal proteins TYR, GPR143, GPNMB, and DCT in the melanosome fraction from indicated cells. (J) Western blot analysis for signature small EV proteins CD9, Syntenin, and CD63 in small EV fractions from indicated cells.

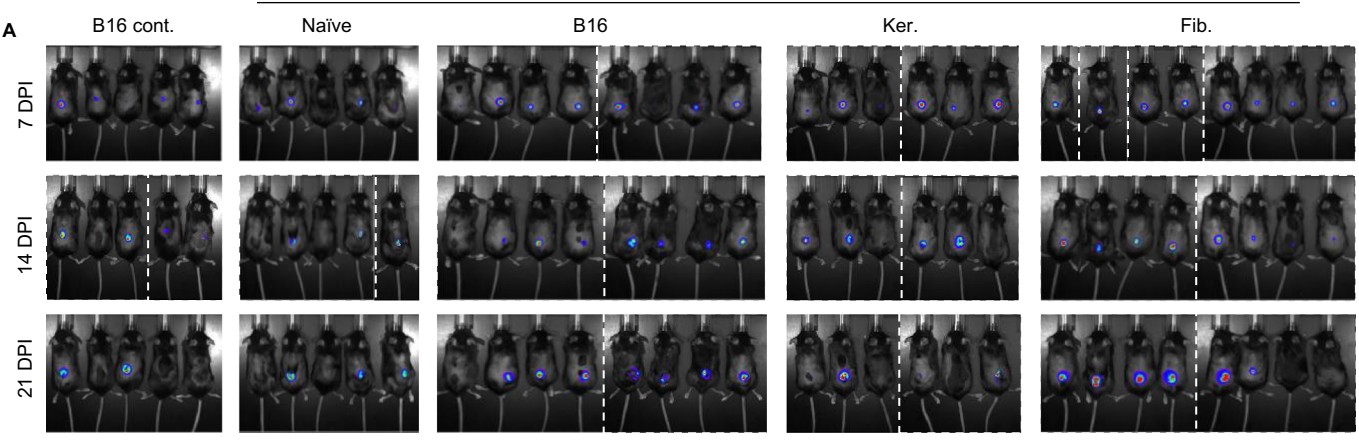

**A** Macrophages fed with melanosomes from-

B16 cont. | Naïve | B16 | Ker. | Fib.

7 DPI

14 DPI

21 DPI

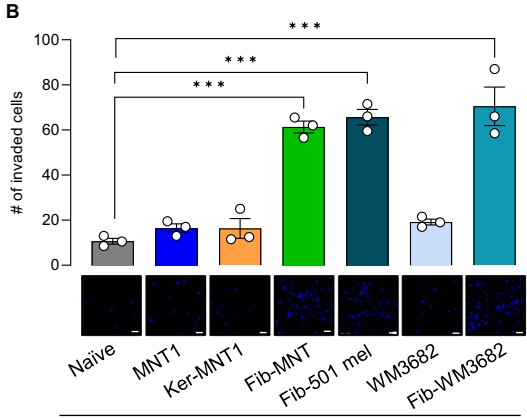

**B**

501 mel cells co-cultured with
CM of naïve macrophages
or macrophages with melanosomes from -

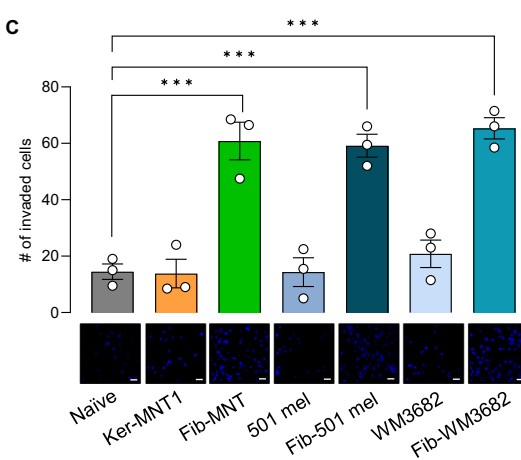

**C**

MNT1 cells co-cultured with
CM of naïve macrophages
or macrophages with melanosomes from -

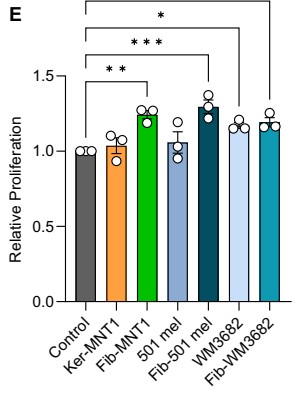

**D**

501 mel cells co-cultured with
CM of naïve macrophages
or macrophages with
melanosomes from -

**E**

MNT1 cells co-cultured with
CM of naïve macrophages
or macrophages with
melanosomes from -

**Figure EV3.  Associated with Fig. 3: Macrophages cultured with melanosomes from different cell sources have heterogenous effects on cancer hallmarks.**

(A) In vivo bioluminescent images of C57BL6 mice injected with B16-F10 mCherry cells expressing firefly luciferase reporter co-inoculated with BMDMs incubated with melanosomes derived from B16-F10 melanoma cells, fibroblasts, or primary keratinocytes. Images were taken on days 7, 14, and 21 post injection. Mice inoculated with B16-F10 cells and naïve macrophages were used as controls. Note: The representative images shown in Fig. 3B haven been taken as a representative of each group from the raw images shown here. (B) Upper: Numbers of invaded 501mel melanoma cells treated with conditioned media from macrophages cultured with melanosomes from indicated source cells for 24 h. $n = 3$ independent experiments. Lower: Representative images of invasion assay. Scale bar, 50 µm. (C) The proliferation of 501mel melanoma cells treated with the conditioned media of macrophages that had been cultured with melanosomes from indicated cells for 24 h relative to culture with conditioned media from naïve macrophages (control). $n = 3$ independent experiments. (D) Upper: Numbers of invaded MNT1 melanoma cells treated with conditioned media from macrophages cultured with melanosomes from indicated source cells for 24 h. $n = 3$ independent experiments. Lower: Representative images of invasion assay. Scale bar, 50 µm. (E) The proliferation of MNT1 melanoma cells treated with the conditioned media of macrophages that had been cultured with melanosomes from indicated cells for 24 h relative to culture with conditioned media from naïve macrophages (control). n = 3 independent experiments. Data information: In panel (B–E), error bars represent ±SEM and one-way ANOVA was performed for statistical analysis, *$P \leq 0.05$, **$P \leq 0.01$, ***$P \leq 0.001$ was considered significant.

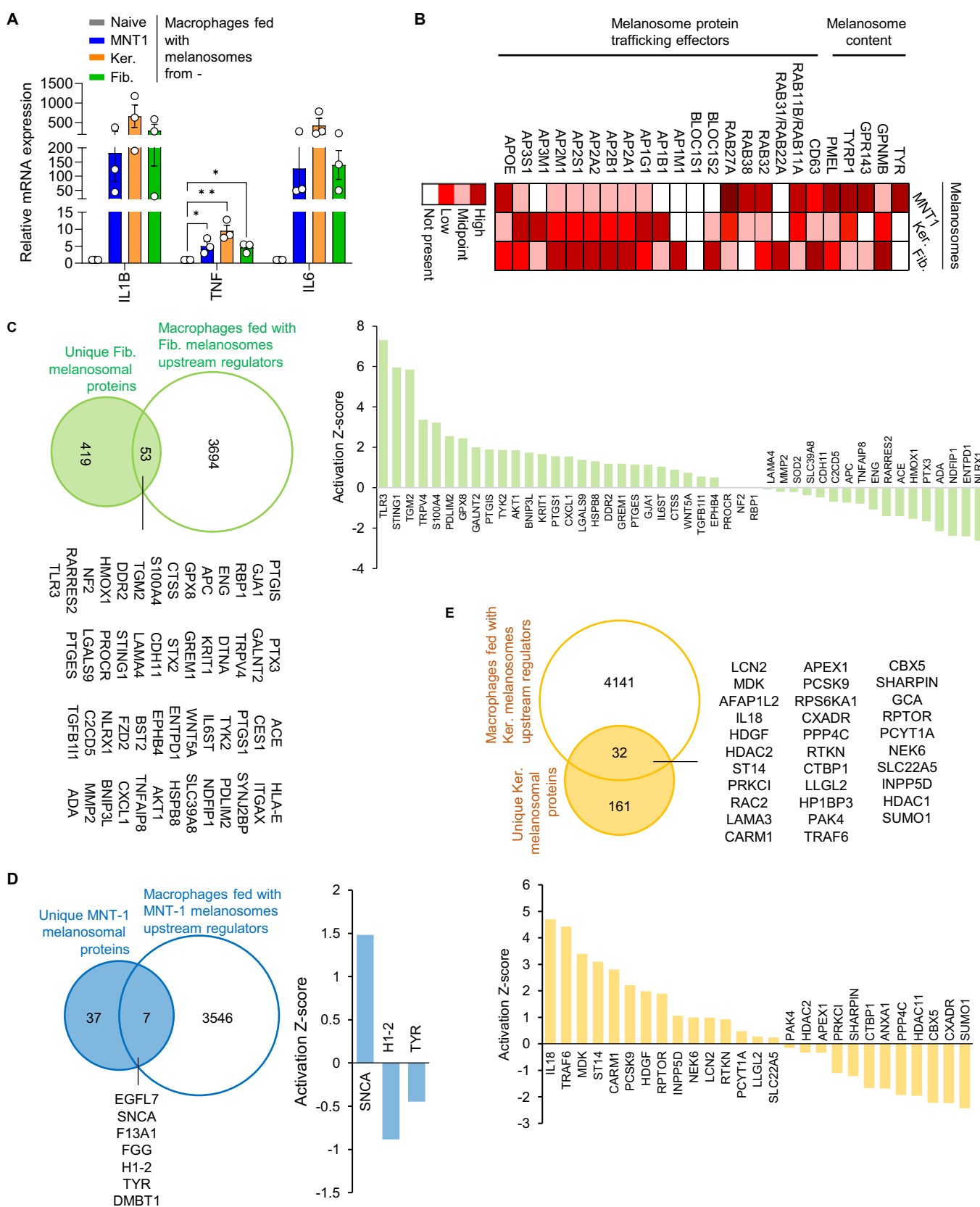

◀

**Figure EV4. Associated with Fig. 4: Macrophages incubated with melanosomes from different cell types are phenotypically and functionally diverse.**

(A) Relative mRNA expression of pro-inflammatory genes *IL1B*, *TNF*, and *IL6* in macrophages cultured with melanosomes from MNT1 melanoma cells, keratinocytes, and fibroblast cells normalized to expression in naïve macrophages; $n = 3$. (B) Heatmap of LFQ intensity of melanosomal proteins and trafficking effector proteins in melanosomes from indicated cell sources. $n = 3$ independent experiments. (C–E) Overlap between the upstream regulators predicted by IPA and unique melanosomal proteins and graphs depicting the activation Z-scores of the upstream regulators found in the overlap from (C) fibroblasts, (D) MNT1 melanoma cells, and (E) keratinocyte as identified in Fig. 4I,J. $n = 3$ individual donors for each melanophage and $n = 3$ independent experiments for melanosome isolation. Data information: In panel (A), error bars represent ±SEM. One-way ANOVA was performed for statistical analysis, *$P \leq 0.05$, **$P \leq 0.01$ was considered significant.

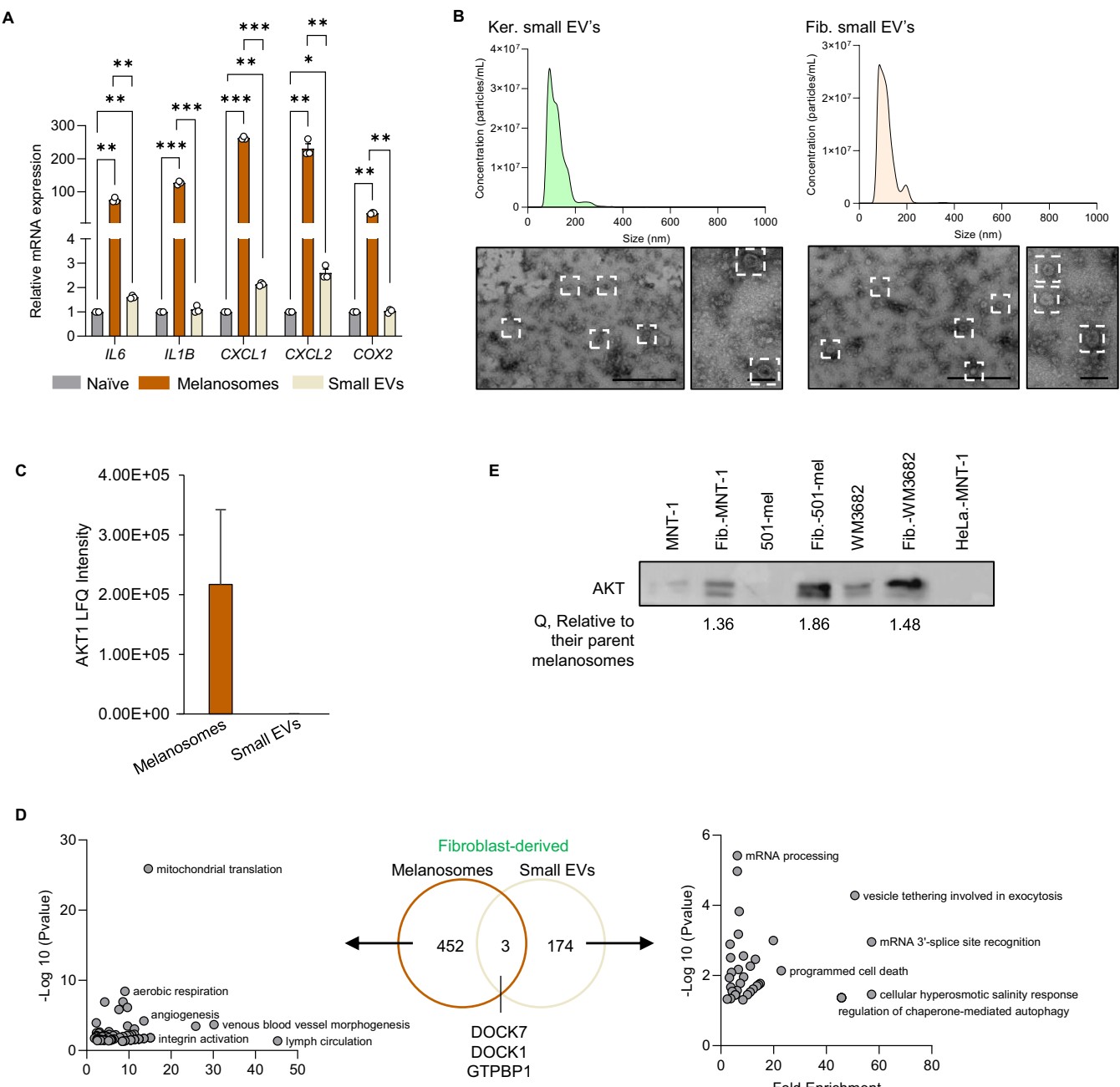

**Figure EV5. Associated with Fig. 5: Fibroblasts melanosomes loaded with AKT1 induce angiogenesis in macrophages via the AKT1/mTOR pathway.**

(A) mRNA levels of CAF-associated genes in dermal fibroblast cells upon culturing with small EVs or melanosomes. $n = 3$ independent experiments. (B) NanoSight analysis (upper) and TEM analysis (lower) of small EVs secreted from keratinocytes (Ker., left panel) and fibroblasts (Fib., right panel). Scale bars, 0.5 µm and 0.2 µm. Small EVs are highlighted with white dashed boxes. (C) LFQ intensity scores of the AKT1 protein levels in fibroblast-derived small EVs and melanosomes. $n = 2$ independent experiments of isolation of each EVs. (D) Center: Overlap between the uniquely expressed proteins in each fibroblast-derived small EVs or melanosomes compared to the MNT1 or keratinocyte-derived small EVs or melanosomes, by proteomics analysis. MNT1 and keratinocyte EVs were used as controls to identify specific proteins expressed by the EVs of the fibroblast cells. Left and right: Dot plots showing the significantly enriched biological processes of the genes further uniquely expressed by each EV. $n = 2$ independent experiments of isolation for each EVs. (E) Western blot analysis for AKT protein levels in melanosomal fractions isolated from the melanoma cells or fibroblast cells that were cultured with indicated melanoma melanosomes. Data information: In panel (A, C), error bars represent ± S.E.M. In panel (A), one-way ANOVA was performed for statistical analysis, *$P \leq 0.05$, **$P \leq 0.01$, ***$P \leq 0.001$ was considered significant.

