## [Peer Review File · The EMBO Journal]

Second-Hand Melanoma Melanosomes Regulate Diversity Among Tumor-Associated Macrophages

Carmit Levy, Roma Parikh, Shivang Parikh, Daniella Berzin, Hananya Vaknine, Shai Ovadia, Daniela Likonen, Shoshana Greenberger, Alon Scope, Sharona Elgavish, Yuval Nevo, Inbar Plaschkes, Eran Nizri, Oren Kobiler, Avishai Maliah, Laureen Zaremba, Vishnu Mohan, Irit Sagi, Ruth Ashery-Padan, Yaron Carmi, Chen Luxenburg, Jörg Hoheisel, Mehdi Khaled, and Mitchell Levesque

Corresponding author(s): Carmit Levy (carmitlevy@post.tau.ac.il)

Review Timeline:

Submission Date:	13th Jun 23
Editorial Decision:	25th Jul 23
Revision Received:	26th Oct 23
Editorial Decision:	14th Dec 23
Revision Received:	4th Jan 24
Editorial Decision:	29th Jan 24
Revision Received:	6th Feb 24
Accepted:	26th Mar 24

Editors: Karin Dumstrei and Ioannis Papaioannou

Transaction Report:

Dear Dr. Levy,

Thank you for submitting your manuscript to The EMBO Journal. Your study has now been seen by two referees and their comments are provided below.

As you can see the referees appreciate your findings and support publication here. They raise some relevant points that would strengthen the analysis. Should you be able to address the concerns raised in full then I would like to invite you to submit a revised version. Let me know if there are anything that we should further discuss.

Thank you for the opportunity to consider your work for publication. I look forward to your revision.

with best wishes

Karin

Karin Dumstrei, PhD
Senior Editor
The EMBO Journal

We realize that it is difficult to revise to a specific deadline. In the interest of protecting the conceptual advance provided by the work, we recommend a revision within 3 months (23rd Oct 2023). Please discuss the revision progress ahead of this time with the editor if you require more time to complete the revisions.

As a matter of policy, competing manuscripts published during this period will not negatively impact on our assessment of the conceptual advance presented by your study.

Use the link below to submit your revision:

Referee #1:

The manuscript of Parikh et al., presents data suggesting that extracellular vehicles (EV) control macrophages, which in turn impact melanoma metastasis. A large body of data, often through complex experimental settings, is presented. Authors suggest that "second hand" EV (up taken by fibroblasts and released from them), engulfed by resident macrophages, cause their polarization and promotion of the metastatic niche. AKT1 was identified as key factor that is responsible for these changes. Human patients' data corroborates elevated AKT1 expression with patient poor responsiveness, elevated melanosome markers and disease aggressiveness. This study could be appropriate for publication in the EMBO J, once the following points can be addressed.

- Most work presented relies on a single melanoma cell line used for EV studies- MNT-1. Can authors observations be corroborated in other melanoma cell lines. Is the phenotype described true for all melanomas (or select genetic mutation driven)?
- Along the same lines, a single cell line was used for proliferation analyses - WM3682; can these be corroborated in additional cell lines?
- Quantifications are lacking in many data figures. Examples are Fig 1B, 1D, 1I, 1J, 1K; likewise, validations for genes identified in omics-based analyses is lacking (i.e., Fig 3D, 3K, 3L).
- Why would B16F10 growth in vivo be attenuated after 14 days (Fig 2C)? this raises concerns regarding the way experiments were performed. Decreased tumor volume after 14 days is indicative of tumor rejection by the host, which may have happened due to the co-culturing with the selective macrophages? Details pertaining to the number of cells injected and why was the ratio 5:1 chosen, etc., are lacking.
- Select subset of genes are selected, at times, but the rationale or unbiased assessment are lacking. For example, Fig 2G highlights changes in mRNA of genes implicated in angiogenesis (Cd31, Tie1, etc.,). Were these genes chosen because they were the most impacted genes?
- AKT1 was selected based on STRING assessment, and experiments followed its altered expression in both EV and patient's specimens. Would AKT inhibitors block the phenotype reported here? Can authors link findings presented earlier (Fig 1-3) with AKT? Can authors establish a formal link between key findings outlined in figures 1-3 and patient's data?

Referee #2:

In this manuscript Parikh et al., investigate how melanosomes originating from melanoma cells and up taken by recipient keratinocytes or dermal fibroblasts can be further transferred to macrophages with consequences on melanoma development. Such "educated" macrophages acquire then a pro-tumor or a pro-immune-cell infiltration phenotype.

Comments and concerns:

The authors make interesting analogies between Extracellular Vesicles (EVs) And Melanosomes which can be assimilated to "large EVs". One should just consider and advise the reader that EVs are defined as membrane delimited vesicles released from the plasma membrane (ectosomes/microvesicles) or from the endosomal system, exosomes (after fusion of multivesicular endosomes with the plasma membrane) (They et al., MISEV Guidelines, JEV, 2022; van Niel et al., Nat Rev Mol. Cell, 2022). In the case that Melanosomes undergo exocytosis to be further up taken by recipient cell, the melanosomal contents once released, are devoid of a limiting membrane. The so called "melanocores" that the authors recover from the cell conditioned media are composed of pigment packed on fibrils composed of the protein PMEL/GP100.

This reviewer would recommend

- Changing/ adapting the title, as the title does not immediately maybe misleading for the reader as these are melanosomes/melanocores and not "EVs" delimited by a membrane per se
- State in the introduction why secreted melanosomes/melanocores can be viewed as large "EVs". Certainly because, as EVs they carry cargo and function in intercellular communication to reprogram recipient cells? What are the analogies.
- With this in mind, also unclear for the reader is the rationale to compare melanosomes and "exosomes".. It is not clear immediately for the EV aficionados that melanosomes are EV like ... and as detailed below the comparison should be with small EVs? Rather than endosome derived "exosomes" ...?

Other comments:

- It would be important even essential to better characterize secreted melanosomes/melanocores but also the small EVs that the

authors call exosomes.

One can "state" exosomes once it is possible to assign a biogenetic origin (endosomal system) (for example Verweij, F et al., JCB, 2018).

Here the authors call exosomes small EVs recovered after differential ultracentrifugation.

Although the authors perform mass spectrometry a couple of additional characterization is required

TEM without or if possible CD63 immunolabeling, Western Blot with small EVs markers.(follow MISEV guidelines, JEV, 2022).

The same holds for isolated melanocores as one cannot discard pigment granules still delimited by a membrane (present in the medium from dead /lysed cells).

Additional and major concerns on data interpretation:

- The authors should revise the interpretation of the EMs presented in Figure 1.

- Although the authors show nice labeling with anti PMEL/GP100 antibodies in the EM observations are very misleading: Panel D .. the authors point to melanosomes in keratinocytes, fibroblasts, macrophages.

- In keratinocytes and macrophages, the organelles do not show any morphological features of bona fide melanosomes transferred to recipient cells (Correia et al., JID, 2018, Hurbain et al., JID, 2018 among others). The panel on fibroblasts is not on focus , difficult to interpret. In keratinocytes the pointed organelles show more morphological features with lysosomes (multiple membranes) , in macrophages they are just electron dense..(actually on the Top left .. the organelle that is half cut could correspond to a melanosome with pigment but not the one pointed that seems more like a "cheddar cheese lysosome" Paul Luzio's papers (Bright et al., Curr Biol, 2005) for example or Klumperman and Raposo, Cold Spring Harbor, 2014).

In conclusion, it is a very interesting study that really deserves attention as there are potential implications, but the data and interpretations should be strengthened. These points should be considered before revision.

Referee #1:

The manuscript of Parikh et al., presents data suggesting that extracellular vesicles (EV) control macrophages, which in turn impact melanoma metastasis. A large body of data, often through complex experimental settings, is presented. Authors suggest that "second hand" EV (up taken by fibroblasts and released from them), engulfed by resident macrophages, caused their polarization and promotion of the metastatic niche. AKT1 was identified as key factor that is responsible for these changes. Human patients' data corroborates elevated AKT1 expression with patient poor responsiveness, elevated melanosomes markers and disease aggressiveness. This study could be appropriate for publication in the EMBO J, once the following points can be addressed.

Most work presented relies on a single melanoma cell line used for EV studies-MNT-1. Can authors observations be corroborated in other melanoma cell lines. Is the phenotype described true for all melanomas (or select genetic mutation driven)?

Along the same lines, a single cell line was used for proliferation analyses – WM3682; can these be corroborated in additional cell lines?

We thank the reviewer for raising important and thoughtful points. To corroborate our phenotypic observations, we have now repeated all experiments in additional human melanoma cell lines 501mel, WM3682, and MNT1. Cells from all three lines were phenotypically analysed following treatment with conditioned media from macrophages that were cultured with (1) ortholog melanosomes isolated from the tested melanoma cells; (2) melanosome isolated from the other melanoma cells; (3) melanosome isolated from fibroblasts that were treated with the ortholog melanoma melanosome, or (4) melanosome isolated from fibroblasts that were treated with the other melanoma line melanosomes.

Invasion: There was a significant increase in invasion potential of melanoma cells from all lines upon treatment with conditioned media of macrophages cultured with fibroblast-derived melanosomes (isolated after treatment with melanosomes from the orthologous melanoma line or from another melanoma line). No increase in melanoma invasion potential was observed upon treatment with conditioned media of macrophages cultured with melanoma melanosomes (**new Figure 3M and Figure EV3B and C**).

Proliferation: A significant increase in proliferation was observed in all melanoma cell lines upon treatment with conditioned media of macrophages that were cultured with fibroblast-derived melanosomes (either the ortholog melanoma or other melanoma line melanosomes). No increase in melanoma proliferation was observed upon treatment with conditioned media of macrophages cultured with melanoma melanosomes (**new Figure 3O and Figure EV3D and E**).

Angiogenesis: A significant increase in the tube formation capability of human dermal blood endothelial cells (HDBEC) was observed upon treatment with conditioned media of macrophages that were cultured with fibroblast-derived melanosomes (either the ortholog melanoma or other melanoma line melanosomes). A smaller but significant increase in the tube formation capability of the HDBEC cells was observed upon treatment with conditioned media of macrophages cultured with melanoma melanosomes (**new Figure 3Q**).

Further, the reviewer raised the possibility that our observations might be a driven by genetic mutation. We therefore considered the genetic status of the tested melanoma cell lines: The 501mel line has mutations in *CDKN2A*, *BRAF*, and *PTEN* (Dahl *et al*, 2013; Tanami *et al*, 2004); the WM3682 line has a mutation in *NRAS* (Lovly *et al*, 2012); and the MNT1 line has a mutation in *BRAF* (Vighi *et al*, 2018). Two lines have *BRAF* mutation, however, with additional various driver mutations, and 70-88% of primary melanomas are *BRAF* mutated (Castellani *et al*, 2023). We cannot exclude the possibility that mutations contribute to the abilities of melanoma melanosomes to affect the fibroblasts and macrophages, either directly or via a driver mutation from the same downstream signaling (Oikonomou *et al*, 2014). However, macrophages promoted melanoma aggressiveness whether they were cultured with melanosomes derived from fibroblasts that were treated with orthologous melanosomes or melanosomes from another melanoma line. Moreover, not only melanosomes from fibroblasts treated with MNT1 melanosomes but also those treated with WM3682 or 501mel melanosomes were loaded with the AKT protein (**new Figure EV5E**). This suggests that our idea about "second hand" EV (uptaken melanosomes released by fibroblasts that are engulfed by resident macrophages cause macrophage polarization that in turn promotes formation of the metastatic niche. Even if only *BRAF*-mutated melanoma causes this effect, this mechanism is relevant in 70-88% of melanoma cases.

Quantifications are lacking in many data figures. Examples are Fig 1B, 1D, 1I, 1J, 1K; likewise,

We have now added quantification to all relevant figures (**Revised Figure 1B and Figure 2I, J** which in the original manuscript corresponds to Figure 1B, J, and K).

validations for genes identified in omics-based analyses is lacking (i.e., Fig 3D, 3K, 3L).

Following the reviewer's important comment, we have now performed validation of the omics-based analyses. We performed qRT-PCR analysis of pro-inflammatory genes and genes identified in Figure 6C in the macrophages that were cultured with melanoma-derived, keratinocyte-derived, or fibroblast-derived melanosomes (**new Figure EV4A and EV6C**, n= 3 independent donors of monocytes that were differentiated into macrophages). To clinically validate the genes identified in omics-based analyses, we have analyzed macrophages within fresh biopsies of lymph nodes harboring melanoma compared to tumor-free sentinel lymph node biopsies of the same patient in collaboration with Dr. Nizri from the Sourasky Medical Center. By flow cytometry we analyzed the expression of proteins including TLR3, as a marker for the macrophages that had taken up fibroblast-derived melanosomes, TRAF6, as a marker of macrophages that had taken up keratinocyte-derived melanosomes, and SNCA, as a marker of macrophages that have taken up melanoma-derived melanosomes. We found that macrophages in the tumor-containing lymph nodes expressed significantly higher levels of TLR3 and TRAF6 compared to those in the tumor-free lymph nodes (**new Figure 6F and workflow in New Figure EV6D**). Simultaneously, we found tumor macrophages expressing solely either TLR3, SNCA, or TRAF6 in the tumor-containing lymph, whereas tumor-free lymph macrophages had a significantly higher expression of the combination of these markers (**new Figure 6F**). Clinical validation of AKT1 was presented in the original manuscript (old Figure 6G).

Why would B16F10 growth be attenuated after 14 days (Fig 2C)? this raises concerns regarding the way experiments were performed. Decreased tumor volume after 14 days is indicative of tumor rejection by the host, which may have happened due to the co-culturing with selective macrophages? Details pertaining to the number of cells injected and why was the ratio 5:1 chosen, etc., are lacking.

We apologize for not providing detailed rationale and methodology for this experiment in the original version. As the reviewer suggested, tumour growth is affected by several parameters including the *number* of injected cells, the *formulation* (Matrigel or PBS) used for the injection and the *ratio* between the co-grafted cells. We injected 0.1 million B16F10 cells via the s.c. route into C57BL6 mice (as in (Wang *et al*, 2019; Pozzi *et al*, 2022; Talebian *et al*, 2012) and tracked tumour growth for 3 weeks. We used this number of cells to avoid a dominant effect of melanoma cell growth that might overshadow the subtle effects involving the tumour microenvironment cells. Such a small number of injected cells might generate a strong immune system as the reviewer suggested. However, co-grafting with specific macrophages that were previously cultured with fibroblast melanosomes significantly overcame this repressor effect (Figure 3C), indicating the importance of macrophages in driving tumor progression. Macrophages cultured with melanoma-derived or keratinocyte-derived melanosomes induced a pro-inflammatory microenvironment (Figure 4D and **new figure EV4A**) that did not suppress melanoma growth. This favours our hypothesis that a strong proinflammatory reaction results when a small number of cells are injected. However, this number of injected cells, clearly allowed us to observe the rescue effect of melanoma cell growth when the cells were co-grafted with macrophages that had been cultured with fibroblast-derived melanosomes (Figure 3D).

Formulation is critical as this provides a surface for the cancer cells to grow and enhances engraftment at the tumor inoculation site (Quintana *et al*, 2008). Specifically, angiogenesis is affected in the presence of Matrigel (Kitahara *et al*, 2010). Therefore, to avoid this bias in favour of melanoma cell growth, we formulated the cells in PBS which might have affected the growth of the melanoma cells in the naive condition. We used a ratio of melanoma cells to tumor environment cells of 1:5 as was done previously (Lerner *et al*. 2023) in order to allow the observation of the environmental cell effect without any bias toward melanoma growth. We have added the above rationale and methods to the revised manuscript.

Select subsets of genes are selected, at times, but the rationale or unbiased assessment are lacking. For example, Fig 2G highlights changes in mRNA of genes implicated in angiogenesis (Cd31, Tie1, etc.,). Were these genes chosen because they were the most impacted genes?

We apologize for not providing the rationale behind the selection of the angiogenesis signature genes. The tested genes were *CD31*, *Tie1*, *Ang1*, and *Ang2*. All are well-established pro-angiogenic genes, known for their roles in enhancing vascular integrity and promoting tumor vascular development (Kim *et al*, 2013; La Porta *et al*, 2018; Pieniasek *et al*, 2018). Furthermore, we included *mTOR*, which encodes a component of the PI3K-AKT-mTOR pathway, which is responsible for releasing VEGF, a crucial factor in tumor angiogenesis (Karar & Maity, 2011). mTOR also plays a key role in polarizing macrophages into tumor-associated macrophages (Chen *et al*, 2012). There was a significant increase in *CD31*, *Tie1*, and *mTOR* gene expression specifically in the tumors of the mice co-grafted with melanoma and macrophages that had taken up fibroblast-derived melanosomes, suggesting a heightened pro-tumor and pro-angiogenic activity of these melanophages. The rationale for selection of these genes is explained in the Results section of the revised manuscript.

AKT1 was selected based on STRING assessment, and experiments followed its altered expression in both EV and patient's specimens. Would AKT inhibitors block the phenotype reported here? Can authors link findings presented earlier (Fig 1-3) with AKT? Can authors establish a formal link between key findings outlined in figures 1-3 and patient's data.

We thank the reviewer for this comment, which led us to enhance the mechanism angle of our manuscript. In response to this feedback, we conducted a series of experiments in the presence of a pan-AKT inhibitor, capivasertib, to assess the angiogenic phenotype, invasion capability, and proliferation potential of WM3682 melanoma cells. Treatment of macrophages that had taken up fibroblast-derived melanosomes, which are loaded with AKT, with the AKT inhibitor repressed tube formation in endothelial cells and tumor invasiveness and proliferation of WM3682 melanoma cells (**new Figure 5G, J, and K**). The new data demonstrate that the effect of macrophages on melanoma phenotype is AKT dependent, providing a formal link between data obtained in culture (Figure 1-3) and the clinical data that revealed AKT and its downstream targets enrichment in tumor macrophages compared to non-tumor macrophages of the same patient (**new Figure 6F, EV6C**, and old Figure 6G).

Referee #2:

In this manuscript Parikh et al., investigate how melanosomes originating from melanoma cells and up taken by recipient keratinocytes or dermal fibroblasts can be further transferred to macrophages with consequences on melanoma development. Such "educated" macrophages acquire then a pro-tumor or pro-immune-cell infiltration phenotype.

Comments and concerns:

The authors make interesting analogies between Extracellular Vesicles (EVs) And Melanosomes which can be assimilated to "large EVs". One should just consider and advise the reader that EVs are defined as membrane delimited vesicles released from the plasma membrane (ectosomes/microvesicles) or from the endosomal system, exosomes (after fusion of multivesicular endosomes with plasma membrane) (They et al., MISEV Guidelines, JEV, 2022; van Niel et al., Nat Rev Mol. Cell, 2022). In the case that Melanosomes undergo exocytosis to be further up taken by recipient cell, the melanosomal contents once released, are devoid of limiting membrane. The so called "melanocores" that the authors recover from the cell conditioned media are composed of pigment packed on fibrils composed of protein PMEL/GP100.

We thank the reviewer and agree that it is necessary to define EVs and explain the origins of melanosomes as membrane delimited vesicles released from the plasma membrane (ectosomes/macrovesicles) or from the endosomal system. This is an important point, especially at the context of melanosomes and exosomes. We have now revised the **introduction**, elaborating this concept as follows: "Extracellular vesicles (EVs) refer to a group of membrane-bound vesicles generated by cells through diverse mechanisms including outward budding from the plasma membrane or inward budding of the endosomal membrane (van Niel *et al*, 2022). EVs are key players in cellular communication networks including the communication between melanoma cells and their microenvironment (Gyukity-Sebestyén *et al*, 2019; Lattmann & Levesque, 2022; Dror *et al*, 2016; Mathieu *et al*, 2019). Melanosomes, which are synthesized specifically in melanocytes, originate from the early endosomal system (Raposo & Marks, 2007) and are tissue-specific, membrane-bound organelles of 200-500 nm in size (Hearing, 2005; Raposo & Marks, 2002) that transport melanin pigment to neighboring epidermal keratinocytes as protection against UV-induced DNA damage (Raposo & Marks, 2007; Yamaguchi & Hearing, 2009; Joshi *et al*, 2007). Although melanosomes are secreted out of the cell of origin and meet other

criteria that could allow them to be classified as EVs [according to the Minimal Information for Studies of Extracellular Vesicles (Théry *et al*, 2018)], including their size and inclusion of a cargo of RNA and protein content, they are not classical EVs at this stage as they are secreted without a membrane and thus have been termed melanocores (Théry *et al*, 2018). However, melanocores gain a membrane once inside keratinocytes, resulting in novel membrane-bound organelles (Tarafder *et al*, 2014; Hurbain *et al*, 2018; Benito-Martínez *et al*, 2021). Further, there are additional ways of melanosome secretion including the shedding of large vesicles that contain melanosomes with their membrane (Théry *et al*, 2018; Benito-Martínez *et al*, 2021).” Moreover, for the sake of clarity and to prevent any potential confusion with classical large EVs, we now consistently use the term melanosomes throughout the revised article.

This reviewer would recommend Changing/adapting the title, as the title does not immediately maybe misleading for the reader as these are melanosomes/melanocores and not "EVs" delimited by a membrane per se

As suggested, we have changed the title to "Second-Hand Melanoma Melanosomes Regulate Diversity Among Tumor-Associated Macrophages".

State in the introduction why secreted melanosomes/melanocores can be viewed as large "EVs". Certainly because, as EVs they carry cargo and function in intercellular communication to reprogram recipient cells? What are the analogies.

This is again very important comment. We have now clarified that melanosomes are EV-like secreted organelles rather than classical EVs. Melanosome size, organelles of 200-500 nm, fit the definition of large EVs as given by the Minimal Information for Studies of Extracellular Vesicles (Théry *et al*, 2018). However, as described above and in the revised text, melanosomes are not classical EVs. We have followed the reviewer’s suggestion and have modified the terminology throughout the paper.

With this in mind, also unclear for the reader is the rationale to compare melanosomes and "exosomes".. It is not clear immediately for the EV aficionados that melanosomes are EV like.. and as detailed below the comparison should be with small EVs? Rather than endosome derived "exosomes"…?

We thank the reviewer for the thoughtful feedback.

First, we will replace the term “exosome” with the term “small EVs”. The comparison between melanosomes and small EVs derived from melanoma cells is appropriate because both can transform primary fibroblasts into CAFs [(Dror *et al*, 2016; Hu & Hu, 2019), and Figure EV5A, respectively]. Our comparison was also motivated by the MISEV guidelines that say that a preliminary step in functional studies should involve comparing the activity of small EVs with that of large EVs (Théry *et al*, 2018). Given that the size of melanosomes is typical of a large EV, we used small EVs as a comparison. Importantly, small EVs are a type of vesicle known to be dissolved in the recipient cells (Gurung *et al*, 2021), whereas melanosomes are not. To clarify our rationale, we have made relevant changes in the Results.

Furthermore, we have included additional information in the Discussion section regarding the rationale for comparing melanosomes with small EVs. Chen *et al*. showed elevated levels of PD1 in both exosomes and larger EVs obtained from metastatic melanoma patients, and that exosomes were primarily responsible for an immunosuppressive effect (Chen *et al*, 2018). Moreover, in another study, AKT1 and other kinases were detected in blood EVs from patients with various epithelial tumor types, which has implications for metastatic prostate cancer due to genomic abnormalities in the PI3K pathway, though the distinction between large and small EVs was not made (Minciocchi *et al*, 2017). It is essential to highlight that focusing exclusively on one type of EV may lead to the potential oversight of the most active subtype for the intended function.

Other comments:

It would be important even essential to better characterize secreted melanosomes/melanocores but also the small EVs that the authors call exosomes. One can "state" exosomes once it is possible to assign a biogenetic origin (endosomal system) (for example Verweij, F *et al*., JCB, 2018).

Following the reviewer comment we now use the term “small EVs” rather than “exosomes” throughout the manuscript. We have now added new supplementary data characterizing EVs by western blot analysis with the signature proteins of melanosomes including GPNMB, GPR143, TYR, and DCT (Dror *et al*, 2016; Raposo & Marks, 2007; Giordano *et al*, 2009; Hoashi *et al*, 2010) (**new Figure EV2I**). We have adhered to the MISEV guidelines defining small EVs, selecting Category 1 and 1b proteins, namely CD63 and CD9, both of which are transmembrane or GPI-

anchored proteins, as well as a Category 2a protein, Syntenin, which is a cytosolic protein found in EVs (Théry *et al*, 2018; Verweij *et al*, 2018) for our analysis (**new Figure EV2J**).

We have also conducted transmission electron microscopy to capture images of the morphologies and sizes of small EVs, secreted melanosomes, and intracellular melanosomes, and NanoSight analysis was performed to determine the sizes of these organelles (**new Figure EV2C**, old Figure EV2A, **and new Figure EV5B**). As melanosomes function in melanin pigment synthesis and storage (Raposo & Marks, 2007), we also measured the melanin content of both small EVs and melanosomes. Melanin was detected within both intracellular and extracellular (secreted) MNT1 melanoma-derived melanosomes, keratinocyte-derived melanosomes, and fibroblast-derived melanosomes (Figure EV2H). No melanin was observed in the small EV fractions as expected (Figure EV2H).

Here the authors call exosomes small EVs recovered after differential ultracentrifugation. Although the authors perform mass spectrometry a couple of additional characterization is required.

TEM without or if possible CD63 immunolabeling, Western Blot with small EVs markers. (follow MISEV guidelines, JEV, 2022). The same holds for isolated melanosomes as one cannot discard pigment granules still delimited by a membrane (present in the medium from dead/lysed cells).

In response to the reviewer's comments, we have changed "exosome" to "small EV" throughout the manuscript. Further, we now better characterize small EVs by analyses of their protein content (**new Figure EV2J**), morphology (**new Figure EV2C and EV5B**), and size (**new Figure EV2C and EV5B**). We also evaluated protein content of secreted and intracellular melanosomes/melanocores (**new Figure EV2I**). Size (Figure EV2A and EV2G), morphology (Figure EV2A and EV2G), and melanin content (Figure EV2H) were presented in the original manuscript.

Additional and major concerns on data interpretation:

The authors should revise the interpretation of the Ems presented in Figure 1.

Although the authors show nice labeling with anti PMEL/GP100 antibodies in the EM observations are very misleading: Panel D.. the authors point to melanosomes in keratinocytes, fibroblasts, macrophages.

In keratinocytes and macrophages, the organelles do not show any morphological features of bona fide melanosomes transferred to recipient cells (Correia et al., JID, 2018, Hurbain et al., JID, 2018 among others). The panel on fibroblasts is not on focus, difficult to interpret. In keratinocytes the pointed organelles show more morphological features with lysosomes (multiple membranes), in macrophages they are just electron dense..(actually on the Top left.. the organelle that is half cut could correspond to a melanosome with pigment but not the one pointed that seems more like a "cheddar cheese lysosome" Paul Luzio's papers (Bright et al., Curr Biol, 2005) for example or Klumperman and Raposo, Cold Spring Harbor, 2014).

We would like to thank the reviewer for the valuable input on better characterizing melanosome structures within each cell type. We have now repeated TEM of keratinocytes, fibroblasts, and macrophages that have taken up MNT1 melanoma-derived melanosomes/melanocores (**new Figure 2B**). Additionally, we have repeated the TEM analysis of macrophages that have taken up keratinocyte-derived and fibroblast-derived melanosomes (**new Figure 2G**). In both we observed the typical melanosome morphology as demonstrated earlier (Correia *et al*, 2018; Hurbain *et al*, 2018). Moreover, we demonstrate that there is a clear separation between the melanosome organelle and the lysosomes surrounded by the multi-membrane sheets within the keratinocytes (**new Figure 2B**).

We acknowledge our prior lack of clarity regarding the melanosome structures within macrophage cells. To address this, we now explicitly state that we have observed the presence of potential broken melanin pigment in a multi-membrane lysosome-like compartment within macrophage cells, possibly indicative of potential organelle degradation. To study whether melanoma-secreted melanosomes/melanocores stay intact or are degraded in a cell-dependent manner, we employed two distinct assays. First, as described in the original version of the paper, we incubated keratinocytes, fibroblasts, and macrophages with MNT1 melanoma-derived secreted melanosomes/melanocores labeled with the Pkh67 dye. We tracked the behavior of these organelles over a span of 3 days. Pkh67 was detected within keratinocytes and fibroblasts over the 3 day period, indicating the presence of melanosomes during the entire time frame of the experiment, whereas the signal within macrophages gradually diminished (**Figure 2D**). Second, for this revision, we have performed a new experiment to demonstrate the co-localization of Pkh67-labeled melanosomes/melanocores within acidic lysosomal structures. Keratinocytes,

fibroblasts, and macrophages were incubated with Pkh67-labeled MNT1 melanoma-derived secreted melanosomes/melanocores for 24 hours. Subsequently, these cells were stained with LysoTracker dye to label acidic organelles such as lysosomes and autolysosomes (Raben *et al*, 2009; Correia *et al*, 2018). We also incubated Pkh67-labeled keratinocyte-derived and fibroblast-derived melanosomes with macrophages for 24 hours followed by LysoTracker staining. Analyses at a single-cell resolution showed that nearly 70% of keratinocytes, fibroblasts, and macrophages were positive for both the LysoTracker and Pkh67-labeled melanosomes (**new Figure EV2K-P**). In about 40% of macrophages, there was co-localization between melanosomes and the LysoTracker dye, whereas co-localization was observed in only 2-3% of fibroblasts and keratinocytes (**new Figure 2H**). Taken together, these observations indicate that melanoma-secreted melanosome/melanocores stay intact or are degraded in a cell-dependent manner.

In conclusion, it is a very interesting study that really deserves attention as there are potential implications, but the data and interpretations should be strengthened. These points should be considered before revision.

We appreciate this reviewer's excellent suggestions, and we have done our best to reply to all points raised.

References:

Benito-Martínez S, Salavessa L, Raposo G, Marks MS & Delevoye C (2021) Melanin Transfer and Fate within Keratinocytes in Human Skin Pigmentation. *Integr Comp Biol* 61: 1546–1555

Castellani G, Buccarelli M, Arasi MB, Rossi S, Pisanu ME, Bellenghi M, Lintas C & Tabolacci C (2023) BRAF mutations in melanoma: biological aspects, therapeutic implications, and circulating biomarkers. *Cancers (Basel)* 15

Chen G, Huang AC, Zhang W, Zhang G, Wu M, Xu W, Yu Z, Yang J, Wang B, Sun H, *et al* (2018) Exosomal PD-L1 contributes to immunosuppression and is associated with anti-PD-1 response. *Nature* 560: 382–386

Chen W, Ma T, Shen X, Xia X, Xu G, Bai X & Liang T (2012) Macrophage-induced tumor angiogenesis is regulated by the TSC2-mTOR pathway. *Cancer Res* 72: 1363–1372

Correction: MYC Mediates Large Oncosome-Induced Fibroblast Reprogramming in Prostate Cancer. (2017) *Cancer Res* 77: 3961

Correia MS, Moreiras H, Pereira FJC, Neto MV, Festas TC, Tarafder AK, Ramalho JS, Seabra MC & Barral DC (2018) Melanin transferred to keratinocytes resides in nondegradative endocytic compartments. *J Invest Dermatol* 138: 637–646

Dahl C, Christensen C, Jönsson G, Lorentzen A, Skjødt ML, Borg Å, Pawelec G & Guldberg P (2013) Mutual exclusivity analysis of genetic and epigenetic drivers in melanoma identifies a link between p14 ARF and RAR β signaling. *Mol Cancer Res* 11: 1166–1178

Dror S, Sander L, Schwartz H, Sheinboim D, Barzilai A, Dishon Y, Apcher S, Golan T, Greenberger S, Barshack I, *et al* (2016) Melanoma miRNA trafficking controls tumour primary niche formation. *Nat Cell Biol* 18: 1006–1017

Giordano F, Bonetti C, Surace EM, Marigo V & Raposo G (2009) The ocular albinism type 1 (OA1) G-protein-coupled receptor functions with MART-1 at early stages of melanogenesis to control melanosome identity and composition. *Hum Mol Genet* 18: 4530–4545

Gurung S, Perocheau D, Touramanidou L & Baruteau J (2021) The exosome journey: from biogenesis to uptake and intracellular signalling. *Cell Commun Signal* 19: 47

Gyukity-Sebestyén E, Harmati M, Dobra G, Németh IB, Mihály J, Zvara Á, Hunyadi-Gulyás É, Katona R, Nagy I, Horváth P, *et al* (2019) Melanoma-Derived Exosomes Induce PD-1 Overexpression and Tumor Progression via Mesenchymal Stem Cell Oncogenic Reprogramming. *Front Immunol* 10: 2459

Hearing VJ (2005) Biogenesis of pigment granules: a sensitive way to regulate melanocyte function. *J Dermatol Sci* 37: 3–14

Hoashi T, Sato S, Yamaguchi Y, Passeron T, Tamaki K & Hearing VJ (2010) Glycoprotein nonmetastatic melanoma protein b, a melanocytic cell marker, is a melanosome-specific and proteolytically released protein. *FASEB J* 24: 1616–1629

Hurbain I, Romao M, Sextius P, Bourreau E, Marchal C, Bernerd F, Duval C & Raposo G (2018) Melanosome distribution in keratinocytes in different skin types: melanosome clusters are not degradative organelles. *J Invest Dermatol* 138: 647–656

Hu T & Hu J (2019) Melanoma-derived exosomes induce reprogramming fibroblasts into cancer-associated fibroblasts via Gm26809 delivery. *Cell Cycle* 18: 3085–3094

Joshi PG, Nair N, Begum G, Joshi NB, Sinkar VP & Vora S (2007) Melanocyte-keratinocyte interaction induces calcium signalling and melanin transfer to keratinocytes. *Pigment Cell Res* 20: 380–384

Karar J & Maity A (2011) Pi3k/akt/mTOR pathway in angiogenesis. *Front Mol Neurosci* 4: 51

Kim O-H, Kang G-H, Noh H, Cha J-Y, Lee H-J, Yoon J-H, Mamura M, Nam J-S, Lee DH, Kim YA, *et al* (2013) Proangiogenic TIE2(+)/CD31 (+) macrophages are the predominant population of tumor-associated macrophages infiltrating metastatic lymph nodes. *Mol Cells* 36: 432–438

Kitahara S, Morikawa S, Shimizu K, Abe H & Ezaki T (2010) Alteration of angiogenic patterns on B16BL6 melanoma development promoted in Matrigel. *Med Mol Morphol* 43: 26–36

Lattmann E & Levesque MP (2022) The role of extracellular vesicles in melanoma progression. *Cancers (Basel)* 14

La Porta S, Roth L, Singhal M, Mogler C, Spegg C, Schieb B, Qu X, Adams RH, Baldwin HS, Savant S, *et al* (2018) Endothelial Tie1-mediated angiogenesis and vascular abnormalization promote tumor progression and metastasis. *J Clin Invest* 128: 834–845

Lovly CM, Dahlman KB, Fohn LE, Su Z, Dias-Santagata D, Hicks DJ, Hucks D, Berry E, Terry C, Duke M, *et al* (2012) Routine multiplex mutational profiling of

melanomas enables enrollment in genotype-driven therapeutic trials. *PLoS ONE* 7: e35309

Mathieu M, Martin-Jaular L, Lavieu G & Théry C (2019) Specificities of secretion and uptake of exosomes and other extracellular vesicles for cell-to-cell communication. *Nat Cell Biol* 21: 9–17

Minciacchi VR, Spinelli C, Reis-Sobreiro M, Cavallini L, You S, Zandian M, Li X, Mishra R, Chiarugi P, Adam RM, *et al* (2017) MYC Mediates Large Oncosome-Induced Fibroblast Reprogramming in Prostate Cancer. *Cancer Res* 77: 2306–2317

van Niel G, Carter DRF, Clayton A, Lambert DW, Raposo G & Vader P (2022) Challenges and directions in studying cell-cell communication by extracellular vesicles. *Nat Rev Mol Cell Biol* 23: 369–382

Oikonomou E, Koustas E, Goulielmaki M & Pintzas A (2014) BRAF vs RAS oncogenes: are mutations of the same pathway equal? Differential signalling and therapeutic implications. *Oncotarget* 5: 11752–11777

Pieniasek M, Matkowski R & Donizy P (2018) Macrophages in skin melanoma—the key element in melanomagenesis. *Oncol Lett* 15: 5399–5404

Pozzi S, Scomparin A, Ben-Shushan D, Yeini E, Ofek P, Nahmad AD, Soffer S, Ionescu A, Ruggiero A, Barzel A, *et al* (2022) MCP-1/CCR2 axis inhibition sensitizes the brain microenvironment against melanoma brain metastasis progression. *JCI Insight* 7

Quintana E, Shackleton M, Sabel MS, Fullen DR, Johnson TM & Morrison SJ (2008) Efficient tumour formation by single human melanoma cells. *Nature* 456: 593–598

Raben N, Shea L, Hill V & Plotz P (2009) Chapter 21 Monitoring Autophagy in Lysosomal Storage Disorders. In *Autophagy in Disease and Clinical Applications, Part C* pp 417–449. Elsevier

Raposo G & Marks MS (2002) The dark side of lysosome-related organelles: specialization of the endocytic pathway for melanosome biogenesis. *Traffic* 3: 237–248

Raposo G & Marks MS (2007) Melanosomes--dark organelles enlighten endosomal membrane transport. *Nat Rev Mol Cell Biol* 8: 786–797

Talebian F, Liu J-Q, Liu Z, Khattabi M, He Y, Ganju R & Bai X-F (2012) Melanoma cell expression of CD200 inhibits tumor formation and lung metastasis via inhibition of myeloid cell functions. *PLoS ONE* 7: e31442

Tanami H, Imoto I, Hirasawa A, Yuki Y, Sonoda I, Inoue J, Yasui K, Misawa-Furihata A, Kawakami Y & Inazawa J (2004) Involvement of overexpressed wild-type BRAF in the growth of malignant melanoma cell lines. *Oncogene* 23: 8796–8804

Tarafder AK, Bolasco G, Correia MS, Pereira FJC, Iannone L, Hume AN, Kirkpatrick N, Picardo M, Torrisi MR, Rodrigues IP, *et al* (2014) Rab11b mediates melanin transfer between donor melanocytes and acceptor keratinocytes via coupled exo/endocytosis. *J Invest Dermatol* 134: 1056–1066

Théry C, Witwer KW, Aikawa E, Alcaraz MJ, Anderson JD, Andriantsitohaina R, Antoniou A, Arab T, Archer F, Atkin-Smith GK, *et al* (2018) Minimal information for studies of extracellular vesicles 2018 (MISEV2018): a position statement of the International Society for Extracellular Vesicles and update of the MISEV2014 guidelines. *J Extracell Vesicles* 7: 1535750

Verweij FJ, Bebelman MP, Jimenez CR, Garcia-Vallejo JJ, Janssen H, Neefjes J, Knol JC, de Goeij-de Haas R, Piersma SR, Baglio SR, *et al* (2018) Quantifying exosome secretion from single cells reveals a modulatory role for GPCR signaling. *J Cell Biol* 217: 1129–1142

Vighi E, Rentsch A, Henning P, Comitato A, Hoffmann D, Bertinetti D, Bertolotti E, Schwede F, Herberg FW, Genieser H-G, *et al* (2018) New cGMP analogues restrain proliferation and migration of melanoma cells. *Oncotarget* 9: 5301–5320

Wang Y, Leonard MK, Snyder DE, Fisher ML, Eckert RL & Kaetzel DM (2019) NME1 Drives Expansion of Melanoma Cells with Enhanced Tumor Growth and Metastatic Properties. *Mol Cancer Res* 17: 1665–1674

Yamaguchi Y & Hearing VJ (2009) Physiological factors that regulate skin pigmentation. *Biofactors* 35: 193–199

Dear Dr. Levy,

Thank you for the submission of your revised manuscript to The EMBO Journal. We have now received the comments of the two referees that were asked to re-evaluate your study (included below). As you will see, both are satisfied with the revision, and they acknowledge that their previous concerns have been sufficiently addressed.

From the editorial side, there are a few changes and corrections that we need from you before we can proceed with acceptance of the manuscript:

- Please enter all relevant funding information (including grant numbers) in our online manuscript handling system. It should match exactly the information provided in the Acknowledgements section of your manuscript.
- Please make sure that all deposited datasets will be publicly available at the time of publication. Reviewer access tokens can now be removed from the Data availability section.
- Please change the heading of your "Declaration of Interests" statement to: "Disclosure and competing interests statement".
- The author contributions statement should be removed from the manuscript file. Instead, we now use CRediT to specify the contributions of each author in the journal submission system. Please use the free text box in our online submission portal to provide more detailed descriptions. See also our guide to authors for more information:
<https://www.embopress.org/page/journal/14602075/authorguide#authorshipguidelines> .
- We noticed that there is a callout in the manuscript for Fig. 1I, but there is no such panel in fig. 1. Please correct this.
- Please provide either a "Yes" or a "Not Applicable" answer to each one of the questions in your Author Checklist (you have not answered two of them yet). In the last column of this checklist, only the sections of the manuscript where the relevant information can be found should be listed (the information per se should be included in the main manuscript file).
- The Appendix should have a brief Table of Contents (including page numbers) on its first page. The nomenclature of its contents should be "Appendix Figure S#", "Appendix Table S#" etc. Please also update the respective callouts in the manuscript accordingly. For more information, see our guide to authors:
<https://www.embopress.org/page/journal/14602075/authorguide#expandedview>
- We noticed that the source data for Fig. 2B have been uploaded individually, not together with other files for Fig. 2. Could you please zip together all source data related to Fig. 2?
- Please note that the final dimensions of the synopsis image should be 550 pixels (wide) x 300-600 pixels (height). Would you please consider increasing the font sizes to make sure all text will be legible at the final dimensions?
- EMBO Press papers are accompanied online by:
 - A) a short (2 sentences) summary of findings and their significance, and
 - B) 2-5 short bullet points highlighting the key findings.Please upload this information in a Word file along with your revised manuscript.
- We detected possible re-use of images between:
 - A) Figure 3G & Figure 5G
 - B) Figure 3B and Figure EV3APlease check carefully these figures, correct them if necessary, or -if image re-use is necessary in any of these cases-, please clarify why and state re-use explicitly in the respective figure legends.
- A highlight box needs to be added to Figure EV2 C.
- A highlight box needs to be added to Figure EV5 B.
- Please consider adding a "Data information" section at the end of each figure legend to report information regarding data, statistics, representation (e.g. error bars, scale bars) etc. referring to the figure panels. Please see our guide to authors for more information:
<https://www.embopress.org/page/journal/14602075/authorguide#figureformat>.
- Please define the annotated p values ***/**/* in the legend of figure 2d, i; 5f-k; EV3b-e as appropriate.
- Please indicate the statistical test used for data analysis in the legend of figure 4I.

- Please note that the box plots need to be defined in terms of minima, maxima, centre, bounds of box and whiskers, and percentile in the legend of figure 4g.
- The movies and their callouts in the manuscript should be renamed "Movie EV1" to "Movie EV5". Their legends should be removed from the main manuscript file, and each one (in a separate Word/text file) should be zipped together with its corresponding movie file.
- Main figure legends should be placed after the References in the manuscript.
- There are 6 EV Figures and the EV Figure legends should be included in the manuscript file, after the main Figure legends; the EV figures themselves should be uploaded individually.
- The manuscript sections are in the wrong order. Please follow the order: Title page, Abstract, Keywords, Introduction, Results, Discussion, Materials and Methods, Data availability, Acknowledgements, Disclosure and competing interests statement, References, Figure legends, Expanded View Figure legends.

Please also note that as part of the EMBO publications' Transparent Editorial Process, The EMBO Journal publishes online a Peer Review File along with each accepted manuscript. This File will be published in conjunction with your paper and will include the referee reports, your point-by-point response and all pertinent correspondence relating to the manuscript. You can opt out of this by letting the editorial office know (contact@embojournal.org). If you do opt out, the Peer Review File link will point to the following statement: "No Peer Review File is available with this article, as the authors have chosen not to make the review process public in this case."

We look forward to seeing a final version of your manuscript as soon as possible. Please use this link to submit your revision:
<https://emboj.msubmit.net/cgi-bin/main.plex>

Yours sincerely,

Referee #1:

The revised manuscript well addresses the comments and suggestions raised by the reviewers and substantiates authors conclusions.

Referee #2:

Although I think that the quality of the EM data could be better, the authors have responded to my concerns satisfactorily and I am satisfied with their revision.

The authors addressed the minor editorial issues.

Dear Dr. Levy,

Thank you for submitting your revised manuscript to The EMBO Journal and for addressing the majority of our previous requests. While checking your manuscript and its associated files, we noticed that there is a number of remaining editorial points that have not been (sufficiently) addressed, and we would thus like to invite another re-submission addressing the following:

1. Please provide in the Data Availability section of your manuscript the specific URL for your mass spectrometry dataset deposited in PRIDE.
2. In the last column of the Author Checklist, only the sections of the manuscript where the relevant information can be found should be listed, e.g. Materials and Methods, Figure legends, Data availability section etc. The information per se should be included in the respective sections of the main manuscript file, not in the Checklist. Please correct your Checklist accordingly.
3. You have uploaded a "Resources Table" in a separate file. If you wish to use the structured methods format in your article, please upload your table as a "Reagent Table" file, and note that the methods (in the main manuscript file) should have the heading "Methods and Protocols". Please make sure your reagent table is formatted according to our examples/templates and does not contain headers referring to other publishers. For more information, examples and templates, please see our guide: <https://www.embopress.org/page/journal/14602075/authorguide#textformat>.
4. Please remove the heading from your synopsis image, as well as the empty space from the bottom of the image - the final dimensions should be 550 pixels (width) x 300-600 pixels (height). We would also recommend simplifying the figure legend -and the image itself, if necessary- keeping only the elements that are essential for interpreting the image correctly.
5. Please use black fonts (not red) for the figure panel labels in all Figures.
6. We have previously detected re-use of images between i. Figure 3G & Figure 5G, and ii. Figure 3B and Figure EV3A, and you have clarified in your revised legends that this is intentional. In this case, please mention the re-use of images in both legends of each pair.
7. The height of your Figure EV2A is extremely long. Please consider moving some of the panels to the Appendix (either to an existing or to a new Appendix Figure) so that Figure EV2A fits vertically in a single page, similarly to all other Figures. Please make sure that all Figure callouts are updated accordingly throughout the manuscript.
8. In the "Data information" section of the legend of Figure 5, the significance level 0.05 presumably corresponds to a single asterisk, but this is not stated. Please check and correct it.
9. Please state clearly the statistical test used for data analysis in the legend of figure 4I.

We look forward to receiving the final version of your manuscript as soon as possible. Please use this link to submit your revision: <https://emboj.msubmit.net/cgi-bin/main.plex>

Yours sincerely,

The authors addressed the remaining editorial issues.

Dear Carmit,

I am very pleased to inform you that your manuscript has been accepted for publication in The EMBO Journal.

Yours sincerely,
